# Stable Neural ODE with Lyapunov-Stable Equilibrium Points for Defending Against Adversarial Attacks

**Qiyu Kang**[*]
Continental-NTU Corporate Lab
Nanyang Technological University
50 Nanyang Avenue, 639798, Singapore
kang0080@e.ntu.edu.sg

**Yang Song**[*]
School of Electrical and Electronic Engineering
Nanyang Technological University
50 Nanyang Avenue, 639798, Singapore
songy@ntu.edu.sg

**Qinxu Ding**
School of Business
Singapore University of Social Sciences
463 Clementi Road, 599494, Singapore
qinxuding@suss.edu.sg

**Wee Peng Tay**
School of Electrical and Electronic Engineering
Nanyang Technological University
50 Nanyang Avenue, 639798, Singapore
wptay@ntu.edu.sg

## Abstract

Deep neural networks (DNNs) are well-known to be vulnerable to adversarial attacks, where malicious human-imperceptible perturbations are included in the input to the deep network to fool it into making a wrong classification. Recent studies have demonstrated that neural Ordinary Differential Equations (ODEs) are intrinsically more robust against adversarial attacks compared to vanilla DNNs. In this work, we propose a stable neural ODE with Lyapunov-stable equilibrium points for defending against adversarial attacks (SODEF). By ensuring that the equilibrium points of the ODE solution used as part of SODEF is Lyapunov-stable, the ODE solution for an input with a small perturbation converges to the same solution as the unperturbed input. We provide theoretical results that give insights into the stability of SODEF as well as the choice of regularizers to ensure its stability. Our analysis suggests that our proposed regularizers force the extracted feature points to be within a neighborhood of the Lyapunov-stable equilibrium points of the ODE. SODEF is compatible with many defense methods and can be applied to any neural network's final regressor layer to enhance its stability against adversarial attacks.

## 1 Introduction

Although deep learning has found successful applications in many tasks such as image classification [1, 2], speech recognition [3], and natural language processing [4], the vulnerability of deep learning to adversarial attacks (e.g., see [5]) has limited its real-world applications due to performance and safety concerns in critical applications. Inputs corrupted with human-imperceptible perturbations can easily fool many vanilla deep neural networks (DNNs) into mis-classifying them and thus significantly impact their performance.

Recent studies [6–8] have applied neural Ordinary Differential Equations (ODEs) [9] to defend against adversarial attacks. Some works like [6] have revealed interesting intrinsic properties of

---

[*]First two authors contributed equally to this work.

35th Conference on Neural Information Processing Systems (NeurIPS 2021).

ODEs that make them more stable than conventional convolutional neural networks (CNNs). The paper [6] proposes a time-invariant steady neural ODE (TisODE) using the property that the integral curves from a ODE solution starting from different initial points (inputs) do not intersect and always preserve uniqueness in the solution function space. However, this does not guarantee that small perturbations of the initial point lead to small perturbations of the integral curve output at a later time $T$. The authors thus proposed a regularizer to limit the evolution of the curves by forcing the integrand to be close to zero. However, neither the non-intersecting property nor the steady-state constraint used in TisODE can guarantee robustness against input perturbations since these constraints do not ensure that the inputs are within a neighborhood of Lyapunov-stable equilibrium points. An example is an ODE that serves as an identity mapping is not robust to input perturbations but satisfies all the constraints proposed in [6].

In this paper, our objective is to design a neural ODE such that the features extracted are within a neighborhood of the Lyapunov-stable equilibrium points of the ODE. We first develop a diversity promoting technique applied in the final fully connected (FC) layer to improve the ODE's stability and analyze the reasons why. We then propose a stable neural ODE with Lyapunov-stable equilibrium points to eliminate the effects of perturbations in the input. From linear control theory [10], a linear time-invariant system $\mathrm{d}\mathbf{z}(t)/\mathrm{d}t = \mathbf{A}\mathbf{z}(t)$, where $\mathbf{A}$ is a constant matrix, is exponentially stable if all eigenvalues of $\mathbf{A}$ have negative real parts. Specifically, we propose to force the Jacobian matrix of the ODE used in the neural ODE to have eigenvalues with negative real parts. Instead of directly imposing constraints on the eigenvalues of the matrix, which lead to high computational complexity when the Jacobian matrix is large, we instead add constraints to the matrix elements to implicitly force the real parts of its eigenvalues to be negative.

Our main contributions are summarized as follows:

1. Based on the concept of Lyapunov-stable equilibrium points, we propose a simple yet effective technique to improve the robustness of neural ODE networks by fixing the final FC layer to be a matrix whose rows have unit norm and such that the maximum cosine similarity between any two rows is minimized. Such a FC layer can be constructed off-line.

2. We propose a stable neural ODE for deFending against adversarial attacks (SODEF) to suppress the input perturbations. We derive an optimization formulation for SODEF to force the extracted feature points to be within a neighborhood of the Lyapunov-stable equilibrium points of the SODEF ODE. We provide sufficient conditions for learning a robust feature representation under SODEF.

3. We test SODEF on several widely used datasets MNIST [11], CIFAR-10 and CIFAR-100 [12] under well-known adversarial attacks. We demonstrate that SODEF is robust against adversarial white-box attacks with improvement in classification accuracy of adversarial examples under PGD attack [13] of up to $44.02\%$, $52.54\%$ and $18.91\%$ percentage points compared to another current state-of-the-art neural ODE network TisODE [6] on MNIST, CIFAR-10 and CIFAR-100, respectively. Similar improvements in classification accuracy of adversarial examples of up to $43.69\%$, $52.38\%$ and $18.99\%$ percentage points compared to ODE net [9] are also obtained.

The rest of this paper is organized as follows. We provide essential preliminaries on neural ODE and its stability analysis in Section 2. In Section 3, we present SODEF model architecture and its training method. We show how to maximize the distance between stable equilibrium points of neural ODEs. We propose an optimization and present theoretical results on its stability properties. We summarize experimental results in Section 4 and conclude the paper in Section 5. The proofs for all lemmas and theorems proposed in this paper are given in the supplementary material. We also refer interested readers to the supplementary material for a more detailed account of related works [14–16, 6] and some popular adversarial attacks [17, 13] that are used to verify the robustness of our proposed SODEF. In the paper, we use lowercase boldface characters like $\mathbf{z}$ to denote vectors in $\mathbb{R}^n$, capital boldface characters like $\mathbf{A}$ to denote matrices in $\mathbb{R}^{n \times n}$, and normal characters like $z$ to denote scalars except that the notation $(x, y)$ are normal characters reserved to denote the input and label pairs. A vector $\mathbf{z} \in \mathbb{R}^n$ is represented as $(\mathbf{z}^{(1)}, \mathbf{z}^{(2)}, \ldots, \mathbf{z}^{(n)})$. The $(i, j)$-th element of a matrix $\mathbf{A}$ is $\mathbf{A}_{ij}$ or $[\mathbf{A}]_{ij}$. The Jacobian matrix of a function $f : \mathbb{R}^n \mapsto \mathbb{R}^n$ evaluated at $\mathbf{z}$ is denoted as $\nabla f(\mathbf{z})$. The set of functions $\mathbb{R}^n \mapsto \mathbb{R}^n$ with continuous first derivatives is denoted as $C^1(\mathbb{R}^n, \mathbb{R}^n)$.

## 2 Preliminaries: Neural ODE and Stability

In a neural ODE layer, the relation between the layer input $\mathbf{z}(0)$ and output $\mathbf{z}(T)$ is described as the following differential equation:

$$\frac{\mathrm{d}\mathbf{z}(t)}{\mathrm{d}t} = f_{\boldsymbol{\theta}}(\mathbf{z}(t), t) \tag{1}$$

where $f_{\boldsymbol{\theta}} : \mathbb{R}^n \times [0, \infty) \mapsto \mathbb{R}^n$ denotes the non-linear trainable layers that are parameterized by weights $\boldsymbol{\theta}$ and $\mathbf{z} : [0, \infty) \mapsto \mathbb{R}^n$ represents the $n$-dimensional state of the neural ODE. Neural ODEs are the continuous analog of residual networks where the hidden layers of residual networks can be regarded as discrete-time difference equations $\mathbf{z}(t + 1) = \mathbf{z}(t) + f_{\boldsymbol{\theta}}(\mathbf{z}(t), t)$. In this work, for simplicity, we only consider the time-invariant (autonomous) case $f_{\boldsymbol{\theta}}(\mathbf{z}(t), t) = f_{\boldsymbol{\theta}}(\mathbf{z}(t))$, where the dynamical system does not explicitly depend on $t$. For such non-linear dynamical systems, the following theorem shows that under mild conditions, its behaviour can be studied via linearization near special points called hyperbolic equilibrium points.

**Theorem 1** (Hartman–Grobman Theorem [18]). *Consider a system evolving in time with state $\mathbf{z}(t) \in \mathbb{R}^n$ that satisfies the differential equation $\dfrac{\mathrm{d}\mathbf{z}(t)}{\mathrm{d}t} = f(\mathbf{z}(t))$ for some $f \in C^1(\mathbb{R}^n, \mathbb{R}^n)$, $f(\mathbf{z}) = (f^{(1)}(\mathbf{z}), \ldots, f^{(n)}(\mathbf{z}))$. Suppose the map has a hyperbolic equilibrium state $\mathbf{z}^* \in \mathbb{R}^n$, i.e., $f(\mathbf{z}^*) = 0$ and the Jacobian matrix $\nabla f = [\partial f^{(i)}/\partial \mathbf{z}^{(j)}]_{i,j=1}^n$ of $f$ evaluated at $\mathbf{z} = \mathbf{z}^*$ has no eigenvalue with real part equal to zero. Then there exists a neighbourhood $N_{\mathbf{z}^*}$ of the equilibrium point $\mathbf{z}^*$ and a homeomorphism $g : N_{\mathbf{z}^*} \mapsto \mathbb{R}^n$, such that $g(\mathbf{z}^*) = 0$ and in the neighbourhood $N_{\mathbf{z}^*}$, the flow of $\dfrac{\mathrm{d}\mathbf{z}(t)}{\mathrm{d}t} = f(\mathbf{z}(t))$ is topologically conjugate by the continuous map $\bar{\mathbf{z}}(t) = g(\mathbf{z}(t))$ to the flow of its linearization $\dfrac{\mathrm{d}\bar{\mathbf{z}}(t)}{\mathrm{d}t} = \nabla f(\mathbf{z}^*) \cdot \bar{\mathbf{z}}(t)$.*

The theorem states that when the Jacobian matrix at the zeros of $f$ has no eigenvalue with zero real part, the behaviour of the original dynamical system can be studied using the simpler linearization of the system around those zeros. We next review some definitions and theorems from linear control theory [10].

**Definition 1** (Lyapunov Stability [10]). *The linear time-invariant system $\dfrac{\mathrm{d}\bar{\mathbf{z}}(t)}{\mathrm{d}t} = \mathbf{A}\bar{\mathbf{z}}(t)$ with constant matrix $\mathbf{A}$ is marginally stable or stable in the sense of Lyapunov if every finite initial state $\bar{\mathbf{z}}(0)$ excites a bounded response. It is asymptotically stable if every finite initial state excites a bounded response, which, in addition, approaches $0$ as $t \to \infty$.*

**Theorem 2** (Lyapunov Stability Theorem [10]). *a) The equation $\dfrac{\mathrm{d}\bar{\mathbf{z}}(t)}{\mathrm{d}t} = \mathbf{A}\bar{\mathbf{z}}(t))$ is marginally stable if and only if all eigenvalues of $A$ have zero or negative real parts and those with zero real parts are simple roots of the minimal polynomial of $\mathbf{A}$. b) The equation $\dfrac{\mathrm{d}\bar{\mathbf{z}}(t)}{\mathrm{d}t} = \mathbf{A}\bar{\mathbf{z}}(t)$ is asymptotically stable if and only if all eigenvalues of $A$ have negative real parts.*

In Theorem 1, we say that a hyperbolic equilibrium point is *Lyapunov-stable* if all eigenvalues of the Jacobian matrix evaluated at it have negative real parts. From Theorems 1 and 2, we see that a small perturbation around the Lyapunov-stable equilibrium point $\mathbf{z}(0)$ leads to $\tilde{\mathbf{z}}(t) \to \mathbf{z}(0)$ as $t \to \infty$, i.e., $\exists \delta > 0$ such that for all $\tilde{\mathbf{z}}(0)$ with $\|\mathbf{z}(0) - \tilde{\mathbf{z}}(0)\|_2 < \delta$, we have $\|\tilde{\mathbf{z}}(t) - \mathbf{z}(0)\|_2 \to 0$ as $t \to \infty$, where $\tilde{\mathbf{z}}(t)$ is the ODE solution for the perturbed input $\tilde{\mathbf{z}}(0)$. In the context of neural network adversarial attacks, if the malicious perturbations around the ODE input $\mathbf{z}(0)$ is small, then the output $\mathbf{z}(T)$ for large enough $T$ will not be affected significantly by the perturbation. Consequently, the succeeding network layers after the neural ODE layer can still perform well without being affected by the input perturbation. The perturbation weakening phenomenon around Lyapunov-stable equilibrium points works like a noise filter and acts as a defense against adversarial attacks.

We require the following definition and result in our stability analysis.

**Definition 2** (Strictly diagonally dominant [19]). *Let $\mathbf{A} \in \mathbb{C}^{n \times n}$. We say that $\mathbf{A}$ is strictly diagonally dominant if $|\mathbf{A}_{ii}| > \sum_{j \neq i} |\mathbf{A}_{ij}|$ for all $i = 1, ..., n$.*

**Theorem 3** (Levy–Desplanques theorem [19]). *If $\mathbf{A} \in \mathbb{C}^{n \times n}$ is strictly diagonally dominant and if every main diagonal entry of $A$ is real and negative, then $A$ is non-singular and every eigenvalue of $A$ has negative real part.*

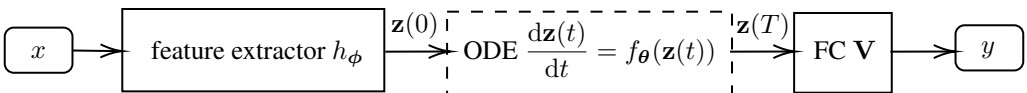

Fig. 1: SODEF model architecture.

**Lemma 1.** *Given $k$ distinct points $\mathbf{z}_i \in \mathbb{R}^n$ and matrices $\mathbf{A}_i \in \mathbb{R}^{n \times n}$, $i = 1, ..., k$, there exists a function $f \in C^1(\mathbb{R}^n, \mathbb{R}^n)$ such that $f(\mathbf{z}_i) = 0$ and $\nabla f_{\boldsymbol{\theta}}(\mathbf{z}_i) = \mathbf{A}_i$.*

## 3 SODEF Architecture

We consider a classification problem with $L$ classes. The proposed SODEF model architecture is shown in Fig. 1. The input $x \in X$ (e.g., an image) is first passed through a feature extractor $h_{\boldsymbol{\phi}} : X \mapsto \mathbb{R}^n$ to obtain an embedding feature representation $\mathbf{z}(0)$. A neural ODE layer $f_{\boldsymbol{\theta}}$ follows as a nonlinear feature mapping to stabilize the feature representation output $\mathbf{z}(0)$ from $h_{\boldsymbol{\phi}}$. The final FC layer $\mathbf{V}$ serves as a linear mapping to generate a prediction vector based on the output $\mathbf{z}(T)$ of the neural ODE layer. The parameters $\boldsymbol{\phi}, \boldsymbol{\theta}$ and $\mathbf{V}$ are parameterized weights for the feature extractor, neural ODE layer and FC layer, respectively.

We provide motivation and design guidance for the FC layer $\mathbf{V}$ in Section 3.1, which attempts to separate Lyapunov-stable equilibrium points implicitly by maximizing the similarity distance between feature representations corresponding to the $L$ different classes. Experimental results demonstrate the advantages of our diversity promoting FC layer in Section 3.1 with comparisons to traditional neural ODEs without diversity promoting.

However, the embedded features after using diversity promoting are not guaranteed to locate near the Lyapunov-stable equilibrium points. In Section 3.2, we formulate an optimization problem to force embedding features to locate near the Lyapunov-stable equilibrium points. We introduce optimization constraints to force the Jacobian matrix of the ODE in our neural ODE layer to have eigenvalues with negative real parts at the Lyapunov-stable equilibrium points. Instead of directly imposing constraints on the eigenvalue of the matrix, which may be computationally complex especially when the matrix is large, we add constraints to the matrix elements instead.

### 3.1 Maximizing the Distance between Lyapunov-Stable Equilibrium Points

From Section 2, we observe that points in a small neighbourhood of a Lyapunov-stable equilibrium point is robust against adversarial perturbations. We call this neighborhood a *stable neighborhood*. However Lyapunov-stable equilibrium points for different classes may very well locate near each other and therefore each stable neighborhood may be very small, leading to poor adversarial defense. In this section, we propose to add a FC layer after the neural ODE layer given by (1) to avoid this scenario. The purpose of the FC layer is to map the output of the neural ODE layer to a feature vector $\mathbf{v}_l$ if the input $x$ belongs to the class $l = 1, \ldots, L$. We design the FC layer so that the cosine similarities between different $\mathbf{v}_l$'s are minimized.

**Lemma 2.** *Given a set of $k$ unit vectors $\mathbf{v}_1, \ldots, \mathbf{v}_k$ in $\mathbb{R}^n$, where $n \geq k$, let $a(\mathbf{v}_1, \ldots, \mathbf{v}_k) = \max_{i \neq j} \mathbf{v}_i^\mathsf{T} \mathbf{v}_j$. Then $\min a(\mathbf{v}_1, \ldots, \mathbf{v}_k) = 1/(1 - k)$, where the minimum is taken over all possible sets of $k$ unit vectors $\mathbf{v}_1, \ldots, \mathbf{v}_k$.*

**Corollary 1.** *Consider a $k \times k$ matrix $\mathbf{B} = [b_{ij}]_{i,j=1}^k$ with $b_{ii} = 1$ and $b_{ij} = 1/(1 - k)$, $\forall i \neq j$. Let the eigen decomposition of $\mathbf{B}$ be $\mathbf{B} = \mathbf{U}\mathbf{\Sigma}\mathbf{U}^\mathsf{T}$. For any $n \geq k$ and $i = 1, \ldots, k$, let $\mathbf{v}_i$ be the $i$-th column of $\mathbf{Q}\mathbf{\Sigma}^{1/2}\mathbf{U}^\mathsf{T}$, where $\mathbf{Q}$ is any $n \times k$ matrix such that $\mathbf{Q}^\mathsf{T}\mathbf{Q} = \mathbf{I}_k$. Then, $a(\mathbf{v}_1, \ldots, \mathbf{v}_k) = \max_{i \neq j} \mathbf{v}_i^\mathsf{T} \mathbf{v}_j = 1/(1 - k)$.*

Corollary 1 suggests a diversity promoting scheme to maximally separate the equilibrium points of the neural ODE layer. The FC layer is represented by an $n \times L$ matrix $\mathbf{V} = [\mathbf{v}_1, \ldots, \mathbf{v}_L]$, where $n$ is the dimension of $\mathbf{z}(T)$, the output from the neural ODE layer. If $\mathbf{z}(T)$ is generated from an input from class $l$, it is mapped to $\mathbf{v}_l$. By minimizing the maximum cosine similarity $a(\mathbf{v}_1, \ldots, \mathbf{v}_k) = \max_{i \neq j} \mathbf{v}_i^\mathsf{T} \mathbf{v}_j$ between the representations from two different classes, we ensure that the output of SODEF is robust to perturbations in the input. Corollary 1 provides a way to choose the FC layer weights $\mathbf{V}$.

To validate our observations, we conduct experiments to compare the robustness of ODE net [9] and TisODE [6] with and without our proposed FC layer $\mathbf{V}$, on two standard datasets: MNIST [2] and CIFAR10 [12] [2]. On the MNIST dataset, all models consist of four convolutional layers and one fully-connected layer. On the CIFAR10 dataset, the networks are similar to those for MNIST except the down-sampling network is a stack of 2 ResNet blocks. In practice, the neural ODE can be solved with different numerical solvers such as the Euler method and the Runge-Kutta methods [9]. Here, we use Runge-Kutta of order 5 in our experiments. Our implementation builds on the open-source neural ODE codes.[3] During training, no Gaussian noise or adversarial examples are augmented into the training set. We test the performance of our model in defending against white-box attacks FGSM [17] and PGD [13] . The parameters for different attack methods used in this paper are given in the supplementary material. From Tables 1 and 2, we observe that for both datasets, our fixed FC layer improves each network's defense ability by a significant margin. We visualize the features before the final FC layer using t-SNE [20] in Figs. 2 and 3. We observe that with the FC layer, the features for different classes are well separated even under attacks.

Table 1: Classification accuracy (%) on adversarial MNIST examples, where the superscript [+] indicates the last FC layer is fixed to be $\mathbf{V}$.

| Attack | Para. | ODE | ODE$^+$ | TisODE | TisODE$^+$ |
|--------|-------|-----|---------|--------|------------|
| None | - | 99.6 | 99.7 | 99.5 | 99.7 |
| FGSM | $\epsilon = 0.3$ | 31.4 | **52.8** | 45.9 | **63.5** |
| PGD | $\epsilon = 0.3$ | 0.29 | **0.30** | 0.4 | **20.20** |

Table 2: Classification accuracy (%) on adversarial CIFAR10 examples, where the superscript [+] indicates the last FC layer is fixed to be $\mathbf{V}$.

| Attack | Para. | ODE | ODE$^+$ | TisODE | TisODE$^+$ |
|--------|-------|-----|---------|--------|------------|
| None | - | 87.0 | 85.0 | 87.4 | 81.8 |
| FGSM | $\epsilon = 0.1$ | 12.9 | **47.6** | 13.1 | **41.9** |
| PGD | $\epsilon = 0.1$ | 7.8 | **14.7** | 7.4 | **16.2** |

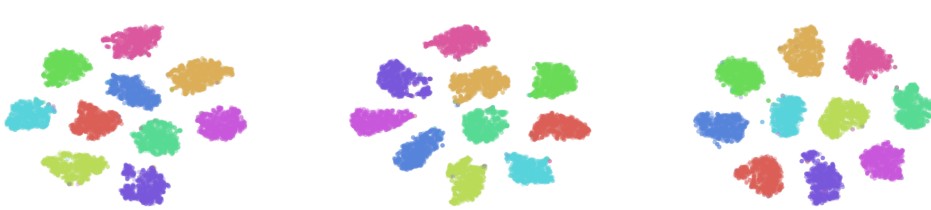

Fig. 2: t-SNE visualization results on the features before the final FC layer. The input is the test set of MNIST. Left: trained with TisODE, middle: TisODE using a randomly chosen orthogonal matrix as the final FC, right: TisODE using proposed $\mathbf{V}$ as the final FC.

## 3.2  Objective Formulation and Stability

In this subsection, we formulate an optimization framework for SODEF to force output features to locate within the stable neighborhood of Lyapunov-stable equilibrium points. We make the following assumption.

**Assumption 1.** *The input $x$ takes values in a compact metric space $X$ and has probability distribution $\mu$. The feature extractor $h_\phi$ is injective and continuous.*

---

[2]Our experiments are run on a GeForce RTX 2080 Ti GPU.
[3]`https://github.com/rtqichen/torchdiffeq`

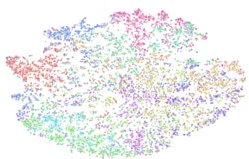 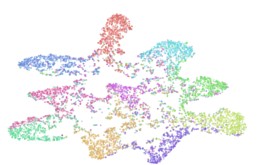 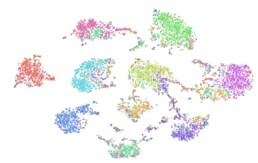

Fig. 3: t-SNE visualization results on the features before the final FC layer. The input is the adversarial examples of the test set of MNIST generated using FGSM method at $\epsilon = 0.3$. Left: trained with TisODE, middle: TisODE using a randomly chosen orthogonal matrix as the final FC, right: TisODE using proposed $\mathbf{V}$ as the final FC.

The above assumption is satisfied if the input $x$ (e.g., an image) resides in a bounded and closed set of a Euclidean space. We denote the pushforward measure (still a probability distribution) of $\mu$ under the continuous feature extractor mapping $h_\phi$ as $\nu_\phi = \mu \circ h_\phi^{-1}$, where $\circ$ denotes function composition. The conditional probability distribution for the embedding of each class $l \in \{1, ..., L\}$ has compact support $E_l \subset \mathbb{R}^n$ since $E_l$ is closed and $h_\phi(X)$ is bounded in $\mathbb{R}^n$. In Section 3.1, the FC layer $\mathbf{V}$ tries to maximize the distance between $E_l$, $l = 1, \ldots, L$. In this section for analysis purposes, we also assume the following.

**Assumption 2.** *We have $E_l \bigcap E_{l'} = \emptyset$ if $l \neq l'$, i.e., the supports of each class are pairwise disjoint.*

Our objective function is formulated as follows, which is explained in detail in the sequel:

$$\min_{\boldsymbol{\theta}, \boldsymbol{\phi}} \mathbb{E}_\mu \ell(\mathbf{V}^\mathsf{T}(\mathbf{z}(T)), y_i) \tag{2}$$

$$\text{s.t. } \mathbb{E}_{\nu_\phi} \|f_{\boldsymbol{\theta}}(\mathbf{z}(0))\|_2 < \epsilon, \ f_{\boldsymbol{\theta}} \in C^1(\mathbb{R}^n, \mathbb{R}^n), \tag{3}$$

$$\mathbb{E}_{\nu_\phi} [\nabla f_{\boldsymbol{\theta}}(\mathbf{z}(0))]_{ii} < 0, \ \forall \, i = 1, \ldots, n, \tag{4}$$

$$\mathbb{E}_{\nu_\phi} \left[ |[\nabla f_{\boldsymbol{\theta}}(\mathbf{z}(0))]_{ii}| - \sum_{j \neq i} |[\nabla f_{\boldsymbol{\theta}}(\mathbf{z}(0))]_{ij}| \right] > 0, \ \forall \, i = 1, \ldots, n, \tag{5}$$

$$\mathbf{z}(0) = h_\phi(x), \text{ and } \mathbf{z}(T) \text{ is the output of (1) with input } \mathbf{z}(0). \tag{6}$$

Here, $\ell$ is a loss function and $\epsilon > 0$ is a positive constant. The constraints (3) to (5) force $\mathbf{z}(0)$ to be near the Lyapunov-stable equilibrium points with strictly diagonally dominant derivatives. We limit the $f_{\boldsymbol{\theta}}$ to be in $C^1(\mathbb{R}^n, \mathbb{R}^n)$ to satisfy the condition in Theorem 1. From [21], we also know that standard multi-layer feed forward networks with as few as a single hidden layer and arbitrary bounded and non-constant activation function are universal approximators for $C^1(\mathbb{R}^n, \mathbb{R}^n)$ functions with respect to some performance criteria provided only that sufficiently many hidden units are available.

As a comparison, TisODE [6] only includes a constraint similar to (3), which in general provides no guarantee to force $\mathbf{z}(0)$ near the Lyapunov-stable equilibrium points. In the extreme case with parameters $\boldsymbol{\theta} = 0$ for $f_{\boldsymbol{\theta}}$ such that $f_{\boldsymbol{\theta}} = 0$, the ODE degenerates to an identity mapping. No $\mathbf{z}(0) \in \mathbb{R}^n$ can now be a Lyapunov-stable equilibrium point, and no stability can therefore be guaranteed to defend against adversarial attacks even though the ODE curves still possess the non-intersecting property and steady-state constraint, which were cited as reasons for the stability of TisODE.

Instead of directly optimizing the above objective function, in our implementation, we optimize the following empirical Lagrangian with a training set $\{(x_k, y_k) : k = 1, ..., N\}$:

$$\min_{\boldsymbol{\theta}, \boldsymbol{\phi}} \frac{1}{N} \sum_{k=0}^{N-1} \left( \ell(\mathbf{V}^\mathsf{T} \mathbf{z}_k(T), y_k) + \alpha_1 \|f_{\boldsymbol{\theta}}(\mathbf{z}_k(0))\|_2 + \alpha_2 g_1 \Big( \sum_{i=1}^n [\nabla f_{\boldsymbol{\theta}}(\mathbf{z}_k(0))]_{ii} \Big) \right.$$

$$\left. + \alpha_3 g_2 \Big( \sum_{i=1}^n (-|[\nabla f_{\boldsymbol{\theta}}(\mathbf{z}_k(0))]_{ii}| + \sum_{j \neq i} |[\nabla f_{\boldsymbol{\theta}}(\mathbf{z}_k(0))]_{ij}|) \Big) \right) \tag{7}$$

$$\text{s.t. } \mathbf{z}_k(0) = h_\phi(x_k), \text{ and } \mathbf{z}_k(T) \text{ is the output of (1) with input } \mathbf{z}_k(0), \ \forall \, k = 1, \ldots, N \tag{8}$$

where $\alpha_1$, $\alpha_2$ and $\alpha_3$ are hyperparameter weights, $g_1$ and $g_2$ are chosen monotonically increasing functions bounded below to eliminate the unbounded impact of the two regularizers that can otherwise dominate the loss. In this paper, we set $g_1(\cdot) = g_2(\cdot) = \exp(\cdot)$. We call these two latter terms the SODEF regularizers.

Suppose for each class $l = 1, \ldots, L$, the embedding feature set $E_l = \{\mathbf{z}_1^{(l)}, \ldots, \mathbf{z}_k^{(l)}\}$ is finite. For each $i = 1, \ldots, k$, let $\mathbf{A}_i \in \mathbb{R}^{n \times n}$ be strictly diagonally dominant matrix with every main diagonal entry be negative such that the eigenvalues for $\mathbf{A}_i$ all have negative real part. From Theorem 3, each $\mathbf{A}_i$ is non-singular and every eigenvalue of $\mathbf{A}_i$ has negative real part. Therefore, from Theorem 2 and Lemma 1, there exists a function $f_{\boldsymbol{\theta}}$ such that all $\mathbf{z}_i^{(l)}$ are Lyapunov-stable equilibrium points with corresponding first derivative $\nabla f_{\boldsymbol{\theta}}(\mathbf{z}_i^{(l)}) = \mathbf{A}_i$. This shows that if there exist only finite representation points for each class, we can find a function $f_{\boldsymbol{\theta}}$ such that all inputs to the neural ODE layer are Lyapunov-stable equilibrium points for $f_{\boldsymbol{\theta}}$ and

(a) $\mathbb{E}_{\nu_{\boldsymbol{\phi}}} \| f_{\boldsymbol{\theta}}(\mathbf{z}(0)) \|_2 = 0$,

(b) $\mathbb{E}_{\nu_{\boldsymbol{\phi}}} [\nabla f_{\boldsymbol{\theta}}(\mathbf{z}(0))]_{ii} < 0$, $\forall i = 1, \ldots, n$,

(c) $\mathbb{E}_{\nu_{\boldsymbol{\phi}}} \left[ |[\nabla f_{\boldsymbol{\theta}}(\mathbf{z}(0))]_{ii}| - \sum_{j \neq i} |[\nabla f_{\boldsymbol{\theta}}(\mathbf{z}(0))]_{ij}| \right] > 0$, $\forall i = 1, \ldots, n$.

If the input space $X$ has infinite cardinality, then an injective and continuous feature extractor $h_{\boldsymbol{\phi}}$ results in a $\nu_{\boldsymbol{\phi}}$ with non-finite support, i.e., at least one $E_l$, $l = 1, \ldots, L$, is infinite. It is not obvious whether we can obtain a $f_{\boldsymbol{\theta}}$ where every point in $E = \bigcup_l E_l$ is a stable equilibrium point. The following result gives a negative answer if $\nu_{\boldsymbol{\phi}}$ is a continuous measure (i.e., absolutely continuous with respect to (w.r.t.) Lebesgue measure) on some subset.

**Lemma 3.** *If the restriction of $\nu_{\boldsymbol{\phi}}$ to some open set $E' \subset E$ is a continuous measure, there is no continuous function $f_{\boldsymbol{\theta}}$ such that for $\nu_{\boldsymbol{\phi}}$-almost surely all $\mathbf{z} \in E$, $f_{\boldsymbol{\theta}}(\mathbf{z}) = 0$ and all the eigenvalues of $\nabla f_{\boldsymbol{\theta}}(\mathbf{z})$ have negative real parts. In other words, there is no continuous function $f_{\boldsymbol{\theta}}$ such that almost surely all $\mathbf{z}$ in $E$ are Lyapunov-stable equilibrium points.*

Lemma 3 indicates that it is too much to hope for all points in $E$ to be Lyapunov-stable equilibrium points. In the following, we relax this requirement and show that under mild conditions, for all $\epsilon > 0$, we can find a continuous function $f_{\boldsymbol{\theta}}$ with finitely many stable equilibrium points such that conditions (b) and (c) above hold and condition (a) is replaced by $\mathbb{E}_{\nu_{\boldsymbol{\phi}}} \| f_{\boldsymbol{\theta}}(\mathbf{z}(0)) \|_2 < \epsilon$. This motivates the optimization constraints in (3) to (5).

**Theorem 4.** *Suppose Assumptions 1 and 2. If $\nu_{\boldsymbol{\phi}}$ is not a continuous uniform measure on $E_l$ for each $l = 1, \ldots, L$, then the following holds: 1) The function space satisfying the constraints in (3) to (5) is non-empty for all $\epsilon > 0$. 2) If additionally the restriction of $\nu_{\boldsymbol{\phi}}$ to any open set $O \subset E_l$ is not a continuous uniform measure, there exist functions in this space such that each support $E_l$ contains at least one Lyapunov-stable equilibrium point.*

## 4 Experiments

In this section, we evaluate the robustness of SODEF under adversarial attacks with different attack parameters. We conduct experiments to compare the robustness of ODE net [9] and TisODE net [6] on three standard datasets: MNIST [2], CIFAR10 and CIFAR100 [3]. Since SODEF is compatible with many defense methods, it can be applied to any neural network's final regressor layer to enhance its stability against adversarial attacks. Our experiment codes are provided in https://github.com/KANGQIYU/SODEF.

### 4.1 Setup

We use open-source pre-trained models that achieve the top accuracy on each dataset as the feature extractor $h_{\boldsymbol{\phi}}$. Specifically for simple MNIST task, we use the ResNet18 model provided in Pytorch. We use the model provided by [22], which obtains nearly $88\%$ clean accuracy on CIFAR100 using EfficientNet [23] and the model provided by [24], which has nearly $95\%$ clean accuracy on CIFAR10. In the neural ODE layer, $f_{\boldsymbol{\theta}}$ consists of 2 FC layers. During the trainings of SODEF (except in the experiment included in Section 4.2), we train the neural network with the fixed FC introduced in Section 3.1. In the first 30 epochs, we fixed $f_{\boldsymbol{\theta}}$ to let the feature extractor $h_{\boldsymbol{\phi}}$ learn a feature representation

with only the cross-entropy loss $\ell$, and in the remaining 120 epochs, we release $h_\phi$ to further train $f_\theta$ using (7) with $\alpha_1 = 1$ and $\alpha_2 = \alpha_3 = 0.05$. For CIFAR10 and CIFAR100, the pixel values are normalized by $(x - \mu)/\sigma$ where $\mu = [0.4914, 0.4822, 0.4465]$ and $\sigma = [0.2023, 0.1994, 0.2010]$ [4]. To show that our SODEF is compatible with many defense methods and can be applied to any neural network's final regression layer, we conduct an experiment where we use a recently proposed robust network TRADES [25] as the feature extractor in our SODEF. The pretrained model is provided here [5], and we choose the model with architecture "WRN-34-10" to conduct our experiments. Besides the two vanilla white-box attacks FGSM and PGD as metioned in Section 3.1, we also include a strong ensemble attack AutoAttack [26], which sequentially performs attack using all of the following four individual attacks: three white-box attacks $\text{APGD}_{\text{CE}}$, $\text{APGD}_{\text{DLR}}^{\text{T}}$ and $\text{FAB}^{\text{T}}$[27], and one black-box Square attack [28]. We refer the reader to the the supplementary material for more details of the attacks used in this paper, where, in additional, more experiments are included.

## 4.2 Compatibility of SODEF

Adversarial training (AT) is one of the most effective strategies for defending adversarial attacks. TRADES [25] is one of the adversarial training defense methods with combinations of tricks of warmup, early stopping, weight decay, batch size and other hyper parameter settings. In this experiment we fix the pretained TRADES model (except the final FC layer (size 640x10)) as our feature extractor $h_\phi$. We then append our (trainable) SODEF with integration time $T = 5$ to the output of the feature extractor. To evaluate model robustness, we use AutoAttack and attack the models using both the $\mathcal{L}_2$ norm ($\epsilon = 0.5$) and $\mathcal{L}_\infty$ norm ($\epsilon = 8/255$). The results are shown in Table 3. We clearly observe that our SODEF can enhance TRADES's robustness under all the four individual attacks and the strongest ensemble AutoAttack. For the strong $\mathcal{L}_2$ AutoAttack, our SODEF have improved the model robustness from $59.42\%$ to $67.75\%$. Our experiment show that SODEF can be applied to many defense models' regression layer to enhance their stability against attacks.

Table 3: Classification accuracy (%) using TRADES (w/ and w/o SODEF) under AutoAttack on adversarial CIFAR10 examples with $\mathcal{L}_2$ norm ($\epsilon = 0.5$) and $\mathcal{L}_\infty$ norm ($\epsilon = 8/255$).

| Attack / Model | TRADES $\mathcal{L}_\infty$ | TRADES+SODEF $\mathcal{L}_\infty$ | TRADES $\mathcal{L}_2$ | TRADES+SODEF $\mathcal{L}_2$ |
|---|---|---|---|---|
| Clean | 85.48 | 85.18 | 85.48 | 85.18 |
| $\text{APGD}_{\text{CE}}$ | 56.08 | **70.90** | 61.74 | **74.35** |
| $\text{APGD}_{\text{DLR}}^{\text{T}}$ | 53.70 | **64.15** | 59.22 | **68.55** |
| $\text{FAB}^{\text{T}}$ | 54.18 | **82.92** | 60.31 | **83.15** |
| Square | 59.12 | **62.21** | 72.65 | **76.02** |
| AutoAttack | 53.69 | **57.76** | 59.42 | **67.75** |

## 4.3 Influence of Integration Time $T$

From the discussion after Theorems 1 and 2, we know if the malicious perturbations around the ODE input Lyapunov-stable equilibrium point $\mathbf{z}(0)$ is small, then the output $\mathbf{z}(T)$ for large enough $T$ will not be affected significantly by the perturbation: $\|\tilde{\mathbf{z}}(t) - \mathbf{z}(0)\|_2 \to 0$ as $t \to \infty$. Consequently, the succeeding network layers after the neural ODE layer can still perform well without being affected by the input perturbation. In this section, we test the influence of the SODEF integration time $T$ using CIFAR100. We use the model EfficientNet provided by [23] as $h_\phi$ (Note, unlike Section 4.2, $h_\phi$ is trainable in this experiments). We use AutoAttack with $\mathcal{L}_2$ norm ($\epsilon = 0.5$). We observe that for all the four individual attacks and the strongest ensemble AutoAttack, SODEF performs generally better for large integration time $T$. We also test larger integration time $T > 10$, but do not see any obvious improvements.

## 4.4 Performance Comparison Under AutoAttack

For a comparison, we provide the results of applying AutoAttack to other baseline models mentioned in the paper. We set the same integration time for ODE, TisODE and SODEF. We observe that for the

---

[4]To test AutoAttack, we have strictly followed the instruction in `https://github.com/RobustBench/robustbench` to attack the original images before any normalization or resizing.

[5]https://github.com/P2333/Bag-of-Tricks-for-AT

Table 4: Classification accuracy (%) under AutoAttack on adversarial CIFAR100 examples with $\mathcal{L}_2$ norm, $\epsilon = 0.5$ and different integration time $T$ for SODEF.

| Attack / $T$ | 1 | 3 | 5 | 6 | 7 | 8 | 9 | 10 |
|---|---|---|---|---|---|---|---|---|
| Clean | 88.00 | 88.12 | 88.15 | 88.00 | 87.92 | 88.00 | 88.05 | 88.10 |
| $\text{APGD}_{\text{CE}}$ | 17.20 | 21.33 | 21.05 | 23.67 | 69.67 | 85.33 | **87.10** | 86.88 |
| $\text{APGD}_{\text{DLR}}^{\text{T}}$ | 21.02 | 21.00 | 22.00 | 26.00 | 63.30 | **86.90** | 86.20 | 86.54 |
| $\text{FAB}^{\text{T}}$ | 86.33 | 85.10 | 86.36 | **87.70** | 87.67 | 86.55 | 86.22 | 85.93 |
| Square | 84.67 | 86.22 | 87.05 | 87.20 | 86.90 | 86.33 | **87.05** | 86.75 |
| AutoAttack | 2.00 | 3.53 | 4.87 | 4.33 | 30.66 | 78.80 | 78.97 | **79.10** |

strongest AutoAttack, our SODEF outperforms the other baseline models by a significant margin. In this case, SODEF achieves 79.10% accuracy while other models only get less than 3% accuracy.

Table 5: Classification accuracy (%) under AutoAttack on adversarial CIFAR100 examples with $\mathcal{L}_2$ norm, $\epsilon = 0.5$ and $T = 10$.

| Attack / Model | No ODE | ODE | TisODE | SODEF |
|---|---|---|---|---|
| Clean | 88.00 | 87.90 | 88.00 | 88.10 |
| $\text{APGD}_{\text{CE}}$ | 23.30 | 6.75 | 14.32 | **86.88** |
| $\text{APGD}_{\text{DLR}}^{\text{T}}$ | 7.33 | 22.00 | 24.20 | **86.54** |
| $\text{FAB}^{\text{T}}$ | 79.30 | 78.67 | 77.16 | **85.93** |
| Square | 84.52 | 85.67 | 86.32 | **86.75** |
| AutoAttack | 0.00 | 1.33 | 4.06 | **79.10** |

## 4.5 Performance Under PGD and FGSM Attacks

White-box adversaries have knowledge of the classifier models, including training data, model architectures and parameters. We test the performance of our model in defending against the white-box attacks, PGD and FGSM. We set $T = 5$ as the integration time for the neural ODE layer. The parameters for different attack methods used are given in the supplementary material. The subsequent experiments use these settings by default, unless otherwise stated.

Table 6: Classification accuracy (%) on adversarial MNIST examples.

| Attack | Para. | no ode | ODE | TisODE | SODEF |
|---|---|---|---|---|---|
| None | - | 99.45 | 99.42 | 99.43 | 99.44 |
| FGSM | $\epsilon = 0.3$ | 10.03 | 29.6 | 36.70 | **63.36** |
| PGD | $\epsilon = 0.3$ | 0.31 | 1.56 | 1.82 | **45.25** |

The classification results on MNIST are shown in Table 6. We observe that while maintaining the state-of-the-art accuracy on normal images, SODEF improves the adversarial robustness as compared to the other two methods. For the most effective attack in this experiment, i.e., PGD attack, SODEF shows a $45.25\% - 1.56\% = 43.69\%$ improvement over ODE and a $45.25\% - 1.23\% = 44.02\%$ improvement over TisODE.

Table 7: Classification accuracy (%) on adversarial CIFAR10 examples.

| Attack | Para. | no ode | ODE | TisODE | SODEF |
|---|---|---|---|---|---|
| None | - | 95.2 | 94.9 | 95.1 | 95.0 |
| FGSM | $\epsilon = 0.1$ | 47.31 | 45.23 | 43.28 | **68.05** |
| PGD | $\epsilon = 0.1$ | 3.09 | 3.21 | 3.80 | **55.59** |

For CIFAR-10, we see from Table 7 that SODEF maintains high accuracy on normal examples and makes the best predictions under adversarial attacks. In particular, SODEF achieves an absolute percentage point improvement over ODE net up to $52.38\%$ and over TisODE up to $52.54\%$ for PGD attack.

For CIFAR-100, the results in the supplementary material shows that the most effective attack causes the classification accuracy to drop relatively by $74.6\% = \frac{88.0 - 22.35}{88.0}$ for SODEF and by

$97.3\% = \frac{88.3 - 2.39}{88.3}$ for vanilla EfficientNet, which is pre-trained on ImageNet to obtain a top clean accuracy. Neither ODE net nor TisODE net can improve the classification accuracy under PGD attack by a big margin, e.g. TisODE net only improves the classification accuracy from 2.39% to 3.44%, while SODEF still shows clear defense capability in this scenario.

### 4.6 Ablation Studies

The impact of the ODE with and without the SODEF regularizers in (7) has been presented in the above comparisons between SODEF and ODE. In this section, we show the necessity of diversity promoting using the FC introduced in Section 3.1 and conduct transferability study.

#### 4.6.1 Impact of Diversity Promotion

Table 8: Classification accuracy (%) on adversarial MNIST examples, where the superscript $^-$ indicates the last FC layer is not fixed to be $\mathbf{V}$ and is set to be a trainable layer.

| Attack | Para. | SODEF | SODEF$^-$ |
|--------|-------|-------|-----------|
| None | - | 95.0 | 95.1 |
| FGSM | $\epsilon = 0.1$ | **63.36** | 51.6 |
| PGD | $\epsilon = 0.1$ | **45.25** | 34.9 |

Table 8 shows the difference of the defense performance when fixing the final FC be $\mathbf{V}$ or setting it to a trainable linear layer. It can be seen that having diversity control improves the robustness. One possible reason for this phenomenon given in Section 3 is that diversity promotion with a fixed designed FC attempts to make the embedding feature support $E_l$ of each class $l$ disjoint to each other and therefore the Lyapunov-stable equilibrium points for each $E_l$ are well separated.

#### 4.6.2 Transferability Study

Transferability study is carried out on CIFAR-10, where the adversarial examples are generated using FGSM and PGD attacks using ResNet18 without any ODEs. The classification accuracy drops from 68.05% to 59% for FGSM with $\epsilon = 0.3$, and from 55.59% to 34% for PGD with $\epsilon = 0.1$. One possible reason for this phenomenon is that ODEs have obfuscated gradient masking effect as discussed in [15], and a transfer attack may deteriorate the defense effect. However, as we observe from Table 7, even with a transfer attack on SODEF, it still performs better than other ODEs without transfer attacks.

## 5 Conclusion

In this paper, we have developed a new neural ODE network, SODEF, to suppress input perturbations. SODEF is compatible with any existing neural networks and can thus be appended to the state-of-the-art networks to increase their robustness to adversarial attacks. We demonstrated empirically and theoretically that the robustness of SODEF mainly derives from its stability and proposed a training method that imposes constraints to ensure all eigenvalues of the Jacobian matrix of the neural ODE layer have negative real parts. When each classification class converges to its own equilibrium points, we showed that the last FC layer can be designed in such a way that the distance between the stable equilibrium points is maximized, which further improves the network's robustness. The effectiveness of SODEF has been verified under several popular while-box attacks.

## Acknowledgments and Disclosure of Funding

This research is supported in part by A*STAR under its RIE2020 Advanced Manufacturing and Engineering (AME) Industry Alignment Fund – Pre Positioning (IAF-PP) (Grant No. A19D6a0053) and the RIE2020 Industry Alignment Fund – Industry Collaboration Projects (IAF-ICP) Funding Initiative, as well as cash and in-kind contribution from the industry partner(s). The computational

work for this article was partially performed on resources of the National Supercomputing Centre, Singapore (https://www.nscc.sg).

## Broader Impact

Our work, which contributes to more robust DNNs, is supposed to mitigate the threat of adversarial attacks. However, on the hand, the reliable deployment of DNNs in automation of tasks will potentially bring mass-scale unemployment and social unrest. As DNNs become more robust and more tasks, especially those whose failures will bring high risks to human lives or large property losses under adversarial attacks, fall into the automatic task category, massive jobs could disappear.

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
