# Supplement

**Qiyu Kang**[*]
Continental-NTU Corporate Lab
Nanyang Technological University
50 Nanyang Avenue, 639798, Singapore
kang0080@e.ntu.edu.sg

**Yang Song**[*]
School of Electrical and Electronic Engineering
Nanyang Technological University
50 Nanyang Avenue, 639798, Singapore
songy@ntu.edu.sg

**Qinxu Ding**
School of Business
Singapore University of Social Sciences
463 Clementi Road, 599494, Singapore
qinxuding@suss.edu.sg

**Wee Peng Tay**
School of Electrical and Electronic Engineering
Nanyang Technological University
50 Nanyang Avenue, 639798, Singapore
wptay@ntu.edu.sg

## 1    Related Work and Adversarial Attacks

In this section, we give an overview of related work in stable neural ODE networks. We also give an overview of common adversarial attacks and recent works that defend against adversarial examples.

**Stable Neural Network**

Gradient vanishing and gradient exploding are two well-known phenomena in deep learning [1]. The gradient of the objective function, which strongly relies on the training method as well as the neural network architecture, indicates how sensitive the output is with respect to (w.r.t.) input perturbation. Exploding gradient implies instability of the output w.r.t. the input and thus resulting in a non-robust learning architecture. On the other hand, vanishing gradient implies insensitivity of the output w.r.t. the input, i.e., robustness against input perturbation. However, this prohibits effective training. To overcome these issues, [2, 3] proposed a new DNN architecture inspired by ODE systems. This DNN architecture is stable in the sense that the input-output of the linearized system is always norm-preserving. This is different from the Lyapunov stability [4] our paper pursues.

The reference [5] developed an implicit Euler based skip-connection to enhance ResNets with better stability and adversarial robustness. More recently, [6] explored the robustness of neural ODE and related the robustness of neural ODE against input perturbation with the non-intersecting property of ODE integral curves. In addition, [6] proposed TisODE, which regularizes the flow on perturbed data via the time-invariant property and the imposition of a steady-state constraint. However, in general, neither the non-intersecting property nor the steady-state constraint can guarantee robustness against input perturbation. In this paper, we propose to impose constraints to ensure Lyapunov stability during training so that adversarial robustness can be enhanced.

**Adversarial Examples**

Recent works [7–9] demonstrated that in image classification tasks, the input images can be modified by an adversary with human-imperceptible perturbations so that a DNN is fooled into misclassifying them. In [10, 11], the authors proposed targeted attacks for audio waveform tasks. They showed that adversarial audio can be embedded into speech so that DNNs cannot recognize the input as human speech. Furthermore, adversarial speech can fool speech-to-text DNN systems into

---

[*]First two authors contributed equally to this work.

35th Conference on Neural Information Processing Systems (NeurIPS 2021).

transcribing the input into any pre-chosen phrase. The references [12, 13] proposed to generate targeted unnoticeable adversarial examples for 3D point clouds, which threatens many safety-critical applications such as autonomous driving.

Adversarial attacks [14–18] can be grouped into two categories: white-box attacks where adversaries have knowledge of the classifier models, including training data, model architectures and parameters, and black-box attacks where adversaries do not know the model's internal architecture or training parameters. An adversary crafts adversarial examples based on a substitute model and then feed these examples to the original model to perform the attack.

To mitigate the effect of adversarial attacks, many defense approaches have been proposed such as adversarial training [7, 14], defensive distillation [19], ECOC based methods [20, 21], and post-training defenses [22–24]. None of these methods make use of the stability of ODEs.

For a normal image-label pair $(x, y)$ and a trained DNN $f_{\boldsymbol{\theta}}$ with $\boldsymbol{\theta}$ being the vector of trainable model parameters, an adversarial attack attempts to find an adversarial example $x'$ that remains within the $\mathcal{L}_p$-ball with radius $\epsilon$ centered at the normal example $x$, i.e., $\|x - x'\|_p \leq \epsilon$, such that $f_{\boldsymbol{\theta}}(x') \neq y$. In what follows, we briefly present some popular adversarial attacks that are used to verify the robustness of our proposed SODEF.

*Fast Gradient Sign Method (FGSM)* [14] perturbs a normal input $x$ in its $\mathcal{L}_\infty$ neighborhood to obtain

$$x' = x + \epsilon \cdot \text{sign}\left(\nabla_x \mathcal{L}(f_{\boldsymbol{\theta}}(x), y)\right),$$

where $\mathcal{L}(f_{\boldsymbol{\theta}}(x), y)$ is the cross-entropy loss of classifying $x$ as label $y$, $\epsilon$ is the perturbation magnitude, and the update direction at each pixel is determined by the sign of the gradient evaluated at this pixel. FGSM is a simple yet fast and powerful attack.

*Projected Gradient Descent (PGD)* [16] iteratively refines FGSM by taking multiple smaller steps $\alpha$ in the direction of the gradient. The refinement at iteration $i$ takes the following form:

$$x'_i = x'_{i-1} + \text{clip}_{\epsilon, x}\left(\alpha \cdot \text{sign}\left(\nabla_x \mathcal{L}(f_{\boldsymbol{\theta}}(x), y)\right)\right),$$

where $x'_0 = x$ and $\text{clip}_{\epsilon, x}(x')$ performs clipping of the input $x'$. For example, in an image $x'$, if a pixel takes values between 0 and 255, $\text{clip}_{\epsilon, x}(x') = \min\{255, x + \epsilon, \max\{0, x - \epsilon, x'\}\}$.

*AutoAttack:* In [25], the authors proposed two extensions of the PGD-attack, named $\text{APGD}_{\text{CE}}$ and $\text{APGD}_{\text{DLR}}^{\text{T}}$, to overcome failures of vanilla PGD due to suboptimal step size and problems of the objective function. They combined these two attacks with two complementary attacks called (white-box) $\text{FAB}^{\text{T}}$[26] and (black-box) Square[27] to form a parameter-free ensemble of attacks, called AutoAttack, to test adversarial robustness.

## 2 Attack Parameters

For two vanilla white-box FGSM and PGD attacks performed in Sections 3.1 and 4, we use $\mathcal{L}_\infty$ norm with maximum perturbation $\epsilon = 0.3$ for MNIST and $\epsilon = 0.1$ for CIFAR10 and we iterates PGD for 20 times with step size 0.1. For the ensemble AutoAttack, we use the standard $\mathcal{L}_2$ norm with maximum perturbation $\epsilon = 0.5$ and $\mathcal{L}_\infty$ norm with maximum perturbation $\epsilon = 8/255$ as in the benchmark[2]

## 3 Influence of Attack Strength

We note that in adversarial attacks, perturbations are assumed to be imperceptible [7], otherwise attack detection techniques can be used [23]. Therefore, most related literature on robust techniques against adversarial attacks assume small perturbation. However, it is hard to obtain a theoretical analysis for an exact perturbation bound, which highly depends on specific datasets and model parameters. We instead demonstrate it experimentally. To show how much perturbation over input $x$ the neural network can defend against, we conduct an experiment here using SODEF where we increase the FGSM attack strength $\epsilon$ at input $x$. Note that in the paper, for CIFAR10 and CIFAR100, the pixel values are normalized by $(x - \mu)/\sigma$ where $\mu = [0.4914, 0.4822, 0.4465]$ and

---

[2]https://robustbench.github.io/

$\sigma = [0.2023, 0.1994, 0.2010]$. The attack results are shown in Fig. 1. The classification accuracy decreases as the attack strength $\epsilon$ increases. As expected, when the perturbation is smaller, SODEF can provide good adversarial robustness.

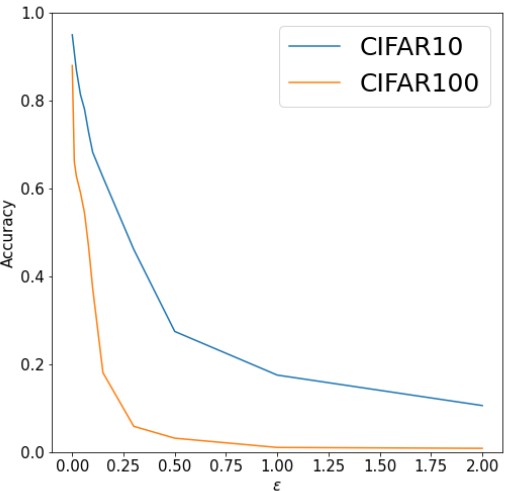

Fig. 1: Classification accuracy under FGSM attack with different parameters $\epsilon$.

Since perturbation in the input $x$ is altered by the feature extractor $h_\phi$, in another experiment, we also directly perturb the feature $\mathbf{z}(0)$ instead of the input $x$. The results are shown in Fig. 2c, where we can observe how the accuracy changes with the attack strength $\epsilon$. Without any neural ODE layer, the classification accuracy decays dramatically and becomes zero when $\epsilon$ is increased to $0.5$. However, while both ODE net and SODEF have demonstrate defense ability over the FGSM attack even when $\epsilon \geq 0.5$, SODEF performs much better than ODE, i.e., the classification accuracy at $\epsilon = 0.5$ for SODEF is $59.9\%$ while ODE net achieves only $18.5\%$ accuracy.

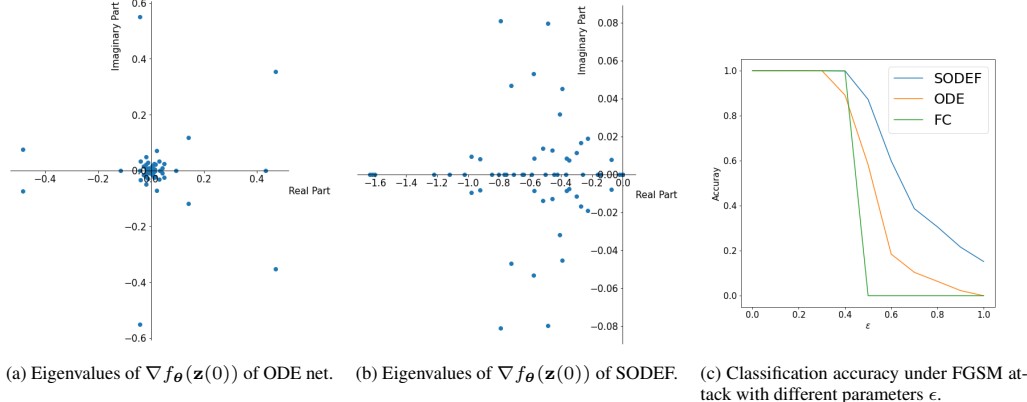

(a) Eigenvalues of $\nabla f_{\boldsymbol{\theta}}(\mathbf{z}(0))$ of ODE net.   (b) Eigenvalues of $\nabla f_{\boldsymbol{\theta}}(\mathbf{z}(0))$ of SODEF.   (c) Classification accuracy under FGSM attack with different parameters $\epsilon$.

Fig. 2: Eigenvalue visualization for the Jacobian matrix $\nabla f_{\boldsymbol{\theta}}(\mathbf{z}(0))$ from ODE net and SODEF, and classification accuracy under FGSM attack with different parameters $\epsilon$.

## 4   Eigenvalues Visualization

To further show the effect of the SODEF regularizers in (7), we compute the eigenvalues of $\nabla f_{\boldsymbol{\theta}}(\mathbf{z}(0))$ from ODE net (no stability constraint is imposed) and SODEF, as shown in Figs. 2a and 2b. We observe that the eigenvalues of the Jacobian matrix from SODEF are located in the left plane, i.e., the eigenvalues all have negative real parts. As a comparison, a large number of ODE net's eigenvalues are located in the right plane, as shown in Fig. 2a.

# 5 Classification Accuracy on CIFAR10 using AutoAttack

In the paper, we study the influence of the SODEF integration time $T$ using CIFAR10. Here in this section, we further test the influence of the SODEF integration time $T$ using CIFAR10. We use the model provided by [28], which has nearly $95\%$ clean accuracy on CIFAR10, as $h_\phi$. We use AutoAttack with $\mathcal{L}_2$ norm ($\epsilon = 0.5$). We still observe that for all the four individual attacks and the strongest ensemble AutoAttack, SODEF has a good performance for large enough integration time $T$.

Table 1: Classification accuracy (%) under AutoAttack on adversarial CIFAR10 examples with $\mathcal{L}_2$ norm, $\epsilon = 0.5$ and different integration time $T$ for SODEF.

| Attack / $T$ | 1 | 3 | 5 | 6 | 7 | 8 | 9 | 10 |
|---|---|---|---|---|---|---|---|---|
| Clean | 95.10 | 94.95 | 95.05 | 95.00 | 94.92 | 95.10 | 95.11 | 95.02 |
| $\text{APGD}_{\text{CE}}$ | 3.52 | 8.20 | 8.59 | 12.59 | 73.05 | 92.41 | 92.58 | **92.67** |
| $\text{APGD}_{\text{DLR}}^{\text{T}}$ | 8.20 | 8.94 | 9.49 | 11.58 | 71.06 | **92.19** | 91.88 | 91.80 |
| $\text{FAB}^{\text{T}}$ | 26.95 | 93.93 | 95.05 | 95.00 | 94.92 | 95.10 | **95.11** | 95.02 |
| Squre | 73.95 | 76.95 | 80.30 | 80.62 | 81.30 | 80.86 | 83.59 | **85.55** |
| AutoAttack | 0.10 | 2.73 | 3.34 | 4.52 | 24.22 | 80.16 | 81.25 | **82.81** |

# 6 Classification Accuracy on CIFAR100 using FGSM and PGD

We further test SODEF on the CIFAR100 dataset. Table 2 shows that SODEF is consistenly more robust than all the competitors, especially when stronger attack is applied.

Table 2: Classification accuracy (%) on adversarial CIFAR100 examples.

| Attack | Para. | no ode | ODE | TisODE | SODEF |
|---|---|---|---|---|---|
| None | - | 88.3 | 88.1 | 88.14 | 88.0 |
| FGSM | $\epsilon = 0.1$ | 25.32 | 29.50 | 31.90 | **37.67** |
| PGD | $\epsilon = 0.1$ | 2.39 | 3.36 | 6.82 | **22.35** |

We then do an experiment for the CIFAR100 dataset where we fix the entire feature extractor (EffcientNet, pretrained on imagenet and CIFAR100, used in our paper already) except for the last layer in the extractor. The results are shown as follows in Table 3. In the first line we use the pretrained EffcientNet with our SODEF and with fine-tuning. In the second line we use a fixed pretrained EffcientNet and with trainable SODEF. In the third line we just attack the pretrained EffcientNet (no ODE and no training since this opensource model has already been trained on CIFAR100 and get $88\%$ clean accuracy).

We observe the fine-tuned model performs the best among three methods and fixed (w/ SODEF) performs better than the vanilla model. One possible reason for fixed (w/ SODEF) being inferior to the fine-tuned model is that the pretrained feature extractor does not have a well diversified $\mathbf{z}(0)$.

Table 3: Classification accuracy (%) using SODEF under PGD and FGSM attack on adversarial CIFAR100 examples with parameters $\epsilon = 0.1$.

| Model / Attack | clean | FGSM | PGD |
|---|---|---|---|
| Fine-tuned (w/ SODEF) | 88.0 | **37.67** | **22.35** |
| Fixed (w/ SODEF) | 87.1 | 33.65 | 17.32 |
| Fixed (no ODE block) | 88.0 | 25.32 | 2.39 |

# 7 Further Ablation Studies

We performed more experiments on the CIFAR10 dataset to better understand the proposed SODEF. We let the feature extractor $h_\phi$ be ResNet18 (which ends up with a feature vector $\mathbf{z}(0) \in \mathbb{R}^{10}$) and

let $f_\theta$ be a linear function such that $f_\theta(\mathbf{z}(t)) = -\mathbf{C}\mathbf{C}^\intercal \mathbf{z}(t)$ where $\theta = \mathbf{C} \in \mathbb{R}^{10 \times 10}$. Since $-\mathbf{C}\mathbf{C}^\intercal$ is a negative definite matrix for full rank $\mathbf{C}$, the ODE block is guaranteed to be asymptotically stable according to Theorem 2. Note that this setup is a special case of our proposed SODEF architecture, which allows $f_\theta$ to be non-linear. The final FC is fixed to be $\mathbf{V}$ given in Corollary 1. For comparison, we construct three other neutral network architectures as described in Table 4.

Table 4: Architectures used in our ablation study.

| Arch. | $h_\phi$ | $f_\theta$ | FC |
|---|---|---|---|
| SODEF | ResNet18 | $-\mathbf{C}\mathbf{C}^\intercal \mathbf{z}(t)$ | $\mathbf{V}$ |
| Net1 | ResNet18 | $-\mathbf{C}\mathbf{C}^\intercal \mathbf{z}(t)$ | random orth. matrix |
| Net2 | ResNet18 | $-\mathbf{C}\mathbf{C}^\intercal \mathbf{z}(t)$ | linear (trainable) |
| Net3 | ResNet18 | $\mathbf{C}^\intercal \mathbf{z}(t) + \mathbf{b}$ | $\mathbf{V}$ |

Table 5 provides the following insights: 1) Comparing SODEF with Net3 highlights the importance of stability of the ODE block; 2) Comparing SODEF with Net 1 and Net2 shows the superiority of our designed FC layer; 3) By comparing Net2 with Net1 and SODEF, we see that enforcing $\mathbf{z}(T)$ to align with a prescribed feature vector, e.g., a row in $\mathbf{V}$, improves model robustness.

Table 5: Classification accuracy (%) on adversarial CIFAR10 examples. We apply PGD for 20 iterations to generate adversarial examples.

| Attack | Para. | SODEF | Net1 | Net2 | Net3 |
|---|---|---|---|---|---|
| None | - | 91.5 | 91.1 | 91.3 | **91.9** |
| PGD | $\epsilon = 0.1$ | **29.2** | 24.7 | 5.6 | 18.8 |

# 8 Further Transferability Studies

We use the same SODEF setup as in Section 7. In order to determine if SODEF's robustness is a result of gradient masking, we generate adversarial examples from two substitute networks whose architectures are given in Table 6 and feed the generated examples into SODEF.

Table 6: Two substitute networks used to generate adversarial examples by running PGD attack with $\epsilon = 0.1$ for 20 iterations. Both networks have no ODE block, i.e., $f_\theta = 0$

| Arch. | $h_\phi$ | $f_\theta$ | FC |
|---|---|---|---|
| NetA | ResNet18 | - | linear (trainable) |
| NetB | ResNet18 | - | $\mathbf{V}$ |

Table 7 provides the following insights: 1) The SODEF column shows that performing the white-box attack directly on SODEF is more effective than performing transferred attacks using substitute models such as NetA and NetB and thereby implying that the ODE block in SODEF does not mask the gradient when performing PGD attack. 2) The NetA and NetB columns show that even when there is no ODE block, NetB using $\mathbf{V}$ as the last FC layer is much more robust than NetA whose FC layer is a trainable dense layer.

Table 7: Classification accuracy (%) on adversarial CIFAR10 examples. We generate adversarial examples by attacking NetA and NetB using PGD method (run for 20 iterations with $\epsilon = 0.1$) and then feed them to SODEF.

| Adv. examples gen. by: | SODEF | NetA | NetB |
|---|---|---|---|
| No Attack | 91.5 | 93.6 | 91.5 |
| SODEF | 29.2 | - | - |
| NetA | 43.9 | 2.6 | - |
| NetB | 51.3 | - | 20.5 |

# 9 Different ODE Solvers

We test different ODE solvers including Runge-Kutta of order 5, Runge-Kutta of order 2, Euler method, Midpoint method, and Fourth-order Runge-Kutta with 3/8 rule. All the ODE solvers tested in Table 8 show similar performance. For Euler method, Midpoint method, and Fourth-order Runge-Kutta with 3/8 rule, we set step size to $0.05$. See `https://github.com/rtqichen/torchdiffeq`.

Table 8: Classification accuracy (%) using SODEF under FGSM attack on adversarial CIFAR10 examples with parameters $\epsilon = 0.1$ and different ODE solvers.

| Model / Attack | clean | FGSM |
|---|---|---|
| Runge-Kutta of order 5 | 95.00 | 68.05 |
| Runge-Kutta of order 2 | 95.12 | 69.95 |
| Euler method | 95.20 | 70.22 |
| Midpoint method | 95.10 | 70.86 |
| Fourth-order Runge-Kutta with 3/8 rule | 95.07 | 69.50 |

# 10 Proofs of Results in Paper

In this section, we provide detailed proofs of the results stated in the main text.

## 10.1 Proof of Lemma 2

For simplicity, let $a = a(\mathbf{v}_1, \ldots, \mathbf{v}_k)$. Since $0 \leq \| \sum_{i=1}^{k} \mathbf{v}_i \|^2 = \sum_i \|\mathbf{v}_i\|^2 + 2\sum_{i \leq j} \mathbf{v}_i^\mathsf{T} \mathbf{v}_j \leq k + 2\sum_{i \leq j} a = k + k(k-1)a$, we have $a \geq 1/(1-k)$. We next prove $a = 1/(1-k)$ is attainable. We can construct a $k \times k$ matrix $\mathbf{B}$ with $\mathbf{B}(i,i) = 1, \forall i$ and $\mathbf{B}(i,j) = 1/(1-k), \forall i \neq j$. It can be verified that $\mathbf{B}$ is a diagonally dominant matrix. Due to the fact that a symmetric diagonally dominant real matrix with nonnegative diagonal entries is positive semi-definite and hence a Gram matrix w.r.t. the Euclidean inner product [29], there exists a set of vectors $\mathbf{v}_i$ such that $\mathbf{v}_i^\mathsf{T} \mathbf{v}_j = \mathbf{B}(i,j) = 1/(1-k)$ for $i \neq j$. In this case, $a(\mathbf{v}_1, \ldots, \mathbf{v}_k) = 1/(1-k)$ and the proof is complete.

## 10.2 Proof of Corollary 1

We may decompose $\mathbf{B}$ as

$$\mathbf{B} = (\mathbf{\Sigma}^{1/2}\mathbf{U}^\mathsf{T})^\mathsf{T}(\mathbf{\Sigma}^{1/2}\mathbf{U}^\mathsf{T})$$
$$= \left(\mathbf{Q}\mathbf{\Sigma}^{1/2}\mathbf{U}^\mathsf{T}\right)^\mathsf{T} \left(\mathbf{Q}\mathbf{\Sigma}^{1/2}\mathbf{U}^\mathsf{T}\right),$$

and hence $\mathbf{B}$ is the Gram matrix of $\{\mathbf{v}_1, \ldots, \mathbf{v}_k\}$. The result follows from the construction of $\mathbf{B}$.

## 10.3 Proof of Lemma 1

The set of finite points $\{\mathbf{z}_1, ..., \mathbf{z}_k\}$ is closed and this lemma is an immediate consequence of the Whitney extension theorem [30].

## 10.4 Proof of Lemma 3

If a continuous function $f$ is such that $f(\mathbf{z}) = 0$ $\nu_\phi$-almost surely, then $f(\mathbf{z}) = 0$ for all $\mathbf{z} \in E'$. Suppose otherwise, then there exists $\mathbf{z}_0$ such that $\|f(\mathbf{z}_0)\|_2 > 0$. From continuity, there exists an open ball $B \in E'$ containing $\mathbf{z}_0$ such that $\|f(\mathbf{z})\|_2 > 0$ for all $\mathbf{z} \in B$. Then since $B$ has non-zero Lebesgue measure, $\nu_\phi(B) > 0$, a contradiction. Therefore, every $\mathbf{z} \in E'$ has $\nabla f(\mathbf{z}) = 0$ and cannot be a Lyapunov-stable equilibrium point.

## 10.5 Proof of Theorem 4

Consider $f(\mathbf{z}) = [f^{(1)}(\mathbf{z}^{(1)}), ..., f^{(n)}(\mathbf{z}^{(n)})]$ with each $f^{(i)}(\mathbf{z}^{(i)}) \in C^1(\mathbb{R}, \mathbb{R})$. Since $f^{(i)}(\mathbf{z}^{(i)})$ only depends on $\mathbf{z}^{(i)}$, $\nabla f_{\boldsymbol{\theta}}(\mathbf{z})$ is a diagonal matrix with all off-diagonal elements being 0. The constraint

(5) is thus satisfied immediately and it suffices to show that there exists such a $f$ satisfying the constraints (3) and (4).

Select a $\mathbf{z}_l = (\mathbf{z}_l^{(1)}, \ldots, \mathbf{z}_l^{(n)})$ from the interior of each $E_l$, $l = 1, \ldots, L$. Let $f^{(i)}(\mathbf{z}^{(i)}) = -\beta(\mathbf{z}^{(i)} - \mathbf{z}_l^{(i)})$ on each $E_l$, where $\beta > 0$. Then $f(\mathbf{z})$ satisfies (4) for all $\beta > 0$ and $\mathbf{z}_l$ is a Lyapunov-stable equilibrium point for each $l$ since $\nabla f_{\boldsymbol{\theta}}(\mathbf{z}_l)$ is a diagonal matrix with negative diagonal values. Since each $E_l \subset \mathbb{R}^n$ is compact, we have that $\forall \epsilon > 0, \exists \beta > 0$ sufficiently small such that $|f^{(i)}(\mathbf{z}^{(i)})| < \epsilon$ for all $\mathbf{z} \in \bigcup_l E_l$. The constraint (3) is therefore satisfied for $f(\mathbf{z})$ with a sufficiently small $\beta$. Since $\bigcup_l E_l$ is closed, the Whitney extension theorem [30] can be applied to extend $f(\mathbf{z})$ to a function in $C^1(\mathbb{R}^n, \mathbb{R}^n)$.