# OpenReview forum: "Stable Neural ODE with Lyapunov-Stable Equilibrium Points for Defending Against Adversarial Attacks"
_NeurIPS.cc/2021/Conference — NeurIPS 2021 Poster_

### Official Review · Reviewer_Bitu · 2021-07-04

**Rating:** 7
**Confidence:** 5

**Summary:**

This work aims to develop a neural ODE-based model to defend against adversarial attacks. Specifically, the authors propose a stable neural ODE with Lyapunov-stable equilibrium points (SODEF). By ensuring the equilibrium points are Lyapunov-stable, the ODE solution for an input with a small perturbation will converge to the same solution for the original unperturbed input. The paper provides theoretical insights into the stability of SODEF and how to choose the stability regularizers. Besides, the paper also provides design guidance for the final FC layer, that aims to separate Lyapunov-stable equilibrium points. The proposed SODEF can work in conjunction with several defense methods and can be applied to any network's final regressor layer.

**Limitations And Societal Impact:**

Hope authors can respond to the questions above.

**Main Review:**

This paper investigates the adversarial robustness from a novel perspective --- stable dynamical system. It is very interesting to study the connection between robust deep networks and the stability of dynamical systems

This paper aims to learn a stable dynamical system-based feature mapping which can mitigate the effect of perturbation. Stable points in a dynamical systems are fixed points with negative-definite Jacobians. Instead of directly compute the eigenvalues of Jacobians, this paper adds constraints to the Jacobian matrices by encouraging the Jacobian to be strictly diagonally dominant. The levy-Desplanques theorem guarantees the stability of systems of which the Jacobians are strictly diagonally dominant and with real negative diagonal entries.

Concern:
1. The formulated optimization problem does not impose regularization on the feature extractor $h_{\phi}$. The optimization-(7) only encourage that the feature extracted by $h_{\phi}$ is located within the neighborhood (say neighborhood with radius $r$) of stable points.  In this case, a perturbation within the $r$-neighborhood on the extracted feature will gradually vanishes.  However, since no regularization on $h_{\phi}$, the imperceptible perturbation on the input $x$ may be amplified --- the perturbed feature is highly likely outside the $r$-neighborhood. Then, the perturbed feature won't converge to the original and the robustness is not guarnteed. So, I think it is necessary to discuss this problem.

2. In section 4.3.2, the authors evaluated the robustness of SODEF with transfer-based attacks. I think it is necessary to evaluate the proposed methods via other more black-box attacks, such as the auto-attack (Croce et al. 2020).

3. The authors provided the implementation codes in the supplementary materials. I skim the codes and find there may be mistakes of the implementation of TisODE. In the file "Tisodenet_cifar10_resnet.py", see lines 461-461, the second term in the loss is  abs (state_t1 - state_t2), which equals to the abs of integral of dynamics from t1 to t2. However, in TisODE (Eq-5), the L_ss is the integral of the abs dynamics from t1 to t2.

4. Some typos. For example, Neural ODE and TisODE are the main baselines in this paper.  In the Reference section, the citation of TisODE is not correct, TisODE is an ICLR 2020 paper, not NeurIPS-2018.

-----------------------------
Reliable Evaluation of Adversarial Robustness with an Ensemble of Diverse Parameter-free Attacks. Francesco Croce,
Matthias Hein. ICML 2020

**Time Spent Reviewing:**

6

---

> ### Author Response · Authors · 2021-08-10
> **We sincerely thank the reviewer for the helpful comments and suggestions! We provide a point-by-point response as follows.**
>
> 1).  We refer the reviewer to our Responses 3 and 4 to Reviewer 1 and Response 4 to Reviewer 3.
>
> Indeed, when the perturbation gets larger as we increase the FGSM attack strength $\epsilon$, we observe the deterioration of robustness as shown in Fig 4c in the paper. In that experiment, we directly perturb the feature $\mathbf{z}(0)$ instead of the input $x$, which shows how much perturbation over feature $\mathbf{z}(0)$ the neural network can defend against.
> Since perturbation in the input $x$ is altered by the feature extractor $h_{ \phi }$, to show how much perturbation over $x$ the neural network can defend against, we have shown additional results in Table 1. in the response to the viewer 1.
>
> We need to emphasize here that we include $\sin$ as our activation function in our ODE block since $\sin$ has many zero equilibrium points and potentially leads to more Lyapunov-stable equilibrium points. This choice makes the ODE solution space full of ''traps'' (Lyapunov-stable equilibrium points) and makes it easier to trap the perturbation. The integration parameter $T$ also influences the defense effect; we again refer the reviewer to Response 4 to Reviewer 3.
>
> Lastly, we need to mention that SODEF is compatible with many defense methods and can be applied to any neural network's final regressor layer to enhance its stability against adversarial attacks. The amplified perturbation mentioned by the reviewer can be weakened by choosing other defense models as our feature extractor $h_{ \phi }$, e.g., TRADES [R4.1].
> In the response to Reviewer 2, we conduct two experiments to show that even if we fix the entire feature extractor (except the last FC layers), the model stability can be improved. In the first experiment, we use a pretrained EffcientNet (see Section 4.1 in our paper) and in the second experiment, we use TRADES [R4.1] to verify SODEF's compatibility and strength. We refer the reviewer to the results shown in Table 2 and Table 3  for details.
>
> [R4.1] Hongyang Zhang, Yaodong Yu, Jiantao Jiao, Eric P Xing, Laurent El Ghaoui, and Michael I Jordan. Theoretically principled trade-off between robustness and accuracy. In International Conference on Machine Learning (ICML), 2019.
>
> 2).  We have evaluated our model using the AutoAttack toolbox. The results are shown in Response 4 to Reviewer 3. We refer the reviewer to the response and experiment results shown in Tables 5-7. We observe that even under AutoAttack, our model still performs well.
>
> Additional results using AutoAttack are shown in Table 3 in the Response 1 to Reviewer 2.
>
> 3).  The second loss is trying to minimize $|z_1(2T)-z_1(T)|$ directly (the leftmost term in eq.(4) in the TisODE paper [R4.2]) instead of its upper bound (the rightmost term in eq.(4) or eq.(5) in [R4.2]). There is no good reason to minimize the upper bound instead and it remains a mystery why the original TisODE paper did so. There is no open source code for TisODE and the regularizer eq.(5) with integration of $|f_{\theta}(z_{i}(t))|$ is not trivial.
> We however also tried to approximate eq.(5) using discretized $L_{ ss }=\left\|\sum_{i=1}^{M}|\int_{T_0}^{T_i}f_{\theta}\left( \mathbf{z} (t)\right) d t|\right\|$,  where $T=T_0< T_1\ldots < T_M=2T$ with $T_i=T+iT/M$ and $M=5$, and observe no obvious change of accuracy when applying attacks. In any case, by our implementation, the effectiveness of TisODE over ODE is preserved especially when strong attacks are performed  (see Tables 7-10 in our response to Reviewer 3). Note that all TisODE results shown in [R4.2] are obtained by augmenting original images together with Gaussian perturbations during training while our implementation of TisODE does not add Gaussian perturbation in training.
>
> [R4.2] H. Yan, J. Du, V. Y. Tan, and J. Feng, “On robustness of neural ordinary differential equations,” Proc. of  International Conference on Learning Representations (ICLR), Addis Ababa, Apr 2020.
>
>
> 4).  We will correct the typos. Thank you.

---

> > ### Comment · Reviewer_Bitu · 2021-08-23
> > **Thanks for the reply**
> >
> > On question (3), the reproduction of TisODE:
> > - The left-most term is eq(4) in the TisODE paper is $\tilde z(T)-z(T)$ but not $z(2T)-z(T)$
> > - As mentioned in the paragraph between eq(4) and eq(5) in the TisODE paper, using $L_{ss}$ is to up-bound the $\tilde z(T)-z(T)$ for all corresponding pairs in set $\mathbb{M}_1$
> >
> > Although the improvement induced by the TisODE baseline may be limited, the authors have to implement all baselines correctly. By the way, I don't think $\sum_i^M |\int_{T_0}^{T_i}fdt|$ is the discretized approximation of $\int_T^{2T} |f|dt$. For example, suppose $f= \sin(wt)$. If $T/M = 2\pi/w$, the $|\int_{T_0}^{T_i}fdt|=0 $, while the $L_{ss}$ is always positive.

---

> > > ### Author Response · Authors · 2021-08-26
> > > **We re-conduct all TisODE experiments using TisODE authors' source code.**
> > >
> > > We appreciate the authors of TisODE providing us the source code so that we can conduct the experiments in exactly the same way as they have done in their paper. In their implementation, a new ODE solver named "FixedGridODESolver_ABS" is defined to obtain the value of the regularizer eq. (5):
> > >
> > > "class Euler_ABS(FixedGridODESolver_ABS):\
> > >     def step_func(self, func, t, dt, y):\
> > >         return tuple(dt * f_ for f_ in func(t, y))\
> > >     def step_func_abs(self, func, t, dt, y):\
> > >         return tuple(dt * torch.abs(f_) for f_ in func(t, y))\
> > >     ...\
> > >         \
> > > class FixedGridODESolver_ABS(object):\
> > >     ...\
> > >     def integrate(self, t):\
> > >         ...\
> > >         for t0, t1 in zip(time_grid[:-1], time_grid[1:]):\
> > >             dy = self.step_func(self.func, t0, t1 - t0, y0)\
> > >             dy_abs = self.step_func_abs(self.func, t0, t1 - t0, y0)\
> > >             y1 = tuple(y0_ + dy_ for y0_, dy_ in zip(y0, dy))\
> > >             y1_abs = tuple(y0_ + dy_ for y0_, dy_ in zip(y0_abs, dy_abs))\
> > >         ...\
> > > "\
> > > We re-conduct __ALL__ TisODE experiments included in the paper, the supplementary materials, and the rebuttal. Specifically, we include the following results: In Table 11, we show new TisODE results for Tables 1-4 in the __paper__. In Table 12, we show new TisODE results for Table 2 in the __supplementary materials__. In Table 13, we show new TisODE results for Tables 7 in the __rebuttal__. In Table 14, we show new TisODE results for Tables 8 and 9 in the __rebuttal__.
> > >
> > > As shown in the tables, we observe that compared to our previous implement, TisODE experiments re-conducted using the authors' source code show only slight improvements in most experiments.  The improvements are limited to several percentage points. In other experiments, it has no improvement. For example, in Table 13, the AutoAttack accuracy on TiSODE improves from $2.52\%$ to $4.06\%$ and the APGD$_{\text{CE}}$ attack accuracy on TiSODE improves from $11.67\%$ to $14.32\%$. SODEF still demonstrates significantly better performance over TisODE in __ALL__ experiments. One reason is that our diversity promoting technique introduced in Section 3.1 leads to a well-diversified $\mathbf{z}(0)$ as shown in Fig. 2 and Fig. 3 in the paper. Another possible reason is well explained in the discussion between eq. (6) and eq. (7) in our paper on page 6:
> > >
> > > "As a comparison, TisODE only includes a constraint similar to (3), which in general provides no guarantee to force $\mathbf{z}(0)$ near the Lyapunov-stable equilibrium points. In the extreme case with parameters ${\mathbf{\theta}}=0$ for $f_{\mathbf{\theta}}$ such that $f_{\mathbf{\theta}}=0$, the ODE degenerates to an identity mapping. No $\mathbf{z}(0)\in\mathbb{R}^n$ can now be a Lyapunov-stable equilibrium point, and no stability can therefore be guaranteed to defend against adversarial attacks even though the ODE curves still possess the non-intersecting property and steady-state constraint, which were cited as reasons for the stability of TisODE."
> > >
> > >
> > > * Additional Discussion:
> > >
> > > I). For a fixed grid $T=T_0<T_1<\dots<T_M=2T$ with $T_i=T_0+iT/M$, the loss integration $\sum_{i=1}^M|\int_{T_{i-1}}^{T_i}f(z(t))dt|$ mentioned in the response 3) is an approximation of the TisODE paper's loss integration $\int_{T}^{2T}|f(z(t))|dt$. We give the demonstration as follows:
> > >
> > > In the fixed step Euler ODE solver in the source code, to obtain eq. (5) $L_{ss}=||\int_{T}^{2 T}|f_{\theta}({z}(t))| d t||$ in the TisODE paper [R4.2], the authors first discretize the integral by $\sum_{i=1}^M\int_{T_{i-1}}^{T_i}|f(z(t))|dt$. For each interval $[T_{i-1}, T_i]$, they use $|f(z(T_{i-1}))|T/M$ to approximate the integral $\int_{T_{i-1}}^{T_i}|f(z(t))|dt$. In our work, we use $|z(T_i)-z(T_{i-1})|$ to approximate the integral $|\int_{T_{i-1}}^{T_i}f(z(t))dt|$. Here, we demonstrate that  $|z(T_i)-z(T_{i-1})|$ is an approximation of  $|f(z(T_{i-1}))|T/M$. For a smooth enough function $f$, using Taylor expansion, we have
> > > \begin{align*}
> > >     z(T_i)= z(T_{i-1})+z'(T_{i-1})T/M+O(1/M^2)= z(T_{i-1})+f(z(T_{i-1}))T/M+O(1/M^2).
> > > \end{align*}
> > > Therefore we have
> > > \begin{align}
> > >    \bigg||z(T_i)-z(T_{i-1})|-|f(z(T_{i-1}))|T/M\bigg|\leq \bigg|z(T_i)-z(T_{i-1})-f(z(T_{i-1}))T/M\bigg|\\ = |O(1/M^2)|.
> > > \end{align}
> > > Thus, the difference between two implementations $\sum_{i=1}^M|z(T_i)-z(T_{i-1})|$ and $\sum_{i=1}^M |f(z(T_{i-1}))|T/M$ is bounded by the term $O(1/M)$ (note the summation over $M$ items here). For a sufficiently large $M$, our implementation is very close to the TisODE paper's implementation. Regarding the $\sin(\omega t)$ example, a sufficiently large $M$ (not the one proposed by the reviewer) for a given $\omega$ gives a good approximation.
> > >
> > >
> > > II). In the TisODE authors' source code, they actually used $L_{\mathrm{ss}}=\sum_{i=1}^{N}\left\|\int_{0.8T}^{1.5T}\left|f_{\theta}\left(\mathbf{z}_{i}(t)\right)\right| \mathrm{d} t\right\|$ instead of the integration from $T$ to $2T$. In all TisODE experiments shown here, we strictly follow their source code.
> > >
> > >
> > >
> > >
> > > |    Tables    |    Table 1     |    Table 2    |     Table 3     |       |     Table 4     |       |     |     |
> > > |:------------:|:--------------:|:-------------:|:---------------:|:-----:|:---------------:|:-----:|:---:|:---:|
> > > | Attack/Model |     TisODE     |    TisODE     |     TisODE      | SODEF |     TisODE      | SODEF |     |     |
> > > |     None     | 99.5 \[99.6\]  | 87.4 \[86.5\] | 99.39 \[99.43\] | 99.44 | 95.12 \[95.0\]  | 95.0  |     |     |
> > > |     FGSM     | 45.9 \[29.10\] | 13.1 \[15.7\] | 36.70 \[31.22\] | 63.36 | 43.28 \[43.58\] | 68.05 |     |     |
> > > |     PGD      | 0.4 \[11.30\]  |  7.4 \[8.0\]  |  1.82 \[1.23\]  | 45.25 |  3.80 \[3.05\]  | 55.59 |     |     |
> > >
> > > __Table 11__ New TisODE experiment results (%) for all tables in the paper. The
> > > original values in the paper are shown in brackets.
> > >
> > >
> > > |    Tables    |    Table 2     |       |
> > > |:------------:|:--------------:|:-----:|
> > > | Attack/Model |     TisODE     | SODEF |
> > > |     None     | 88.14 \[87.9\] | 88.0  |
> > > |     FGSM     | 31.90 \[30.5\] | 37.67 |
> > > |     PGD      | 6.82 \[3.44\]  | 22.35 |
> > >
> > > __Table 12.__ New TisODE experiment results (%) for all tables in the supplementary
> > > material. The original values in the supplementary material are shown in
> > > brackets.
> > >
> > >
> > >
> > >
> > >
> > >
> > > |             Tables             |     Table 7     |       |
> > > |:------------------------------:|:---------------:|:-----:|
> > > |          Attack/Model          |     TisODE      | SODEF |
> > > |             Clean              | 88.04 \[88.0\]  | 88.10 |
> > > |       APGD$_{\text{CE}}$       | 14.32 \[11.67\] | 86.88 |
> > > | APGD$^{\text{T}}_{\text{DLR}}$ | 24.20 \[22.42\] | 86.54 |
> > > |        FAB$^{\text{T}}$        | 77.16 \[79.50\] | 85.93 |
> > > |             Square             | 86.32 \[86.00\] | 86.75 |
> > > |           AutoAttack           |  4.06 \[2.52\]  | 79.10 |
> > >
> > > __Table 13.__ New TisODE experiment results (%) for Table 7 in the rebuttal. The
> > > original values in the rebuttal are shown in brackets.
> > >
> > >
> > >
> > >
> > >
> > >
> > > |       Tables       |                |    Table 8    |       |    Table 9     |       |     |     |     |     |
> > > |:------------------:|:--------------:|:-------------:|:-----:|:--------------:|:-----:|:---:|:---:|:---:|:---:|
> > > |    Attack/Model    |     para.      |    TisODE     | SODEF |     TisODE     | SODEF |     |     |     |     |
> > > |        None        |       \-       | 95.1 \[95.0\] | 95.0  | 88.14 \[87.9\] | 88.0  |     |     |     |     |
> > > |  step size $0.1$   | $\epsilon=0.1$ | 3.80 \[3.05\] | 55.59 | 6.82 \[3.44\]  | 22.35 |     |     |     |     |
> > > | step size $0.0125$ | $\epsilon=0.1$ | 0.96 \[1.03\] | 52.45 | 1.36 \[1.38\]  | 16.35 |     |     |     |     |
> > >
> > > __Table 14.__ New TisODE experiment results (%) for Table 8 and Table 9 in the
> > > rebuttal. The original values in the rebuttal are shown in brackets.

---

> > > > ### Comment · Reviewer_Bitu · 2021-08-26
> > > > **Thanks for the reply.**
> > > >
> > > > I appreciate the authors' effort to reproduce baselines correctly. The authors have addressed my concerns. I am willing to increase my scores. Related experimental results and discussion in this rebuttal session may be included in the later version of the paper.

---

### Official Review · Reviewer_dTd1 · 2021-07-16

**Rating:** 6
**Confidence:** 4

**Summary:**

The paper proposes a defense against adversarial attacks using two orthogonal ideas: (i) the design of the final fully connected layer that maximizes the cosine distance between vectors corresponding to different labels, and (ii) design of a neural ODE whose Jacobian has only negative real parts and is hence Lyapunov stable near its hyperbolic equilibrium points (via Hartman-Grobman and linearization).

**Limitations And Societal Impact:**

(-) The experimental evaluation of the proposed approach, including results in the supplementary material, is limited. A good reference for completing a thorough evaluation is [3]. At the very minimum, PGD attacks with varying L2 and Linf norms should be reported. Modern attacks such as FABS and Square Attack should be evaluated. Can we obtain a report on the results of applying AutoAttack including PGD, FABS and SquareAttacks on these models?

(-) The quality of the reported experimental results is low. For example, Table 1 in Sec. 2 of the Supplementary Material shows that the PGD attack uses a maximum perturbation size of epsilon=0.1 and a step size of 0.1 with 20 iteration steps (not size). The step size should be much smaller such as 2.5*0.1/20 following Madry [4]. AutoAttack https://github.com/fra31/auto-attack provides a parameter-free approach of performing such an evaluation.

(-) The Hartman-Grobman linearization holds only near hyperbolic equilibrium points. Why are these points the only points of interest for robustness of neural networks? Why is the dynamics in the rest of the state space not of interest for robustness analysis?

(-) Figure 3 states that the input is the adversarial examples of the test set of MNIST.  Are these plots obtained by looking at the adversarial examples obtained by attacking each of the three models: TisODE, TisODE and TisODE + V as FC?



Minor points:

1. Reference 8 focuses on stochastic differential equations and not ordinary differential equations. So, paragraph 2 on page 1 may need to be suitably edited.

2. Table 1 has no results for applying PGD to MNIST while Table 2 applies both FGSM and PGD to CIFAR10.

3. Table 3 does not specify the norm used for the PGD/FGSM attack.

4. The paper or the Supplementary Material does not include investigations that sweep the magnitude of the perturbation (epsilon) and observe the impact on the robustness of the model.

References:

[3] Carlini N, Wagner D. Towards evaluating the robustness of neural networks. In 2017 ieee symposium on security and privacy (sp) 2017 May 22 (pp. 39-57). IEEE. https://arxiv.org/abs/1608.04644

[4] https://github.com/MadryLab/robustness



**Main Review:**

(+) While Hartman-Grobman theorem has been used in the stability analysis and generalization/robustness of neural networks via linearization around hyperbolic equilibrium points (see Section 4 of [1] and Section 5.2 of [2]), its use in training adversarially robust neural networks is new.

(+) The conclusion highlights that the approach is compatible with any existing neural network. It is clear that residual neural nets and their variants can benefit from this approach. It may be useful to comment on how this approach can be naturally extended to transformers without residual layers.

(+) The source code has been made available to the reviewers. Can the authors confirm that the complete pipeline in Figure 1 from x to y is being attacked during experimental studies?

[1] Kim P, Pan L, Wirjanto TS. Local Stability Analysis of Neural Network Models with Application to Exchange-Rate Data. Mimeo, University of Waterloo, Waterloo, Ontario, Canada; 1999. https://www.researchgate.net/profile/Tony-Wirjanto/publication/228936073_Local_Stability_Analysis_of_Neural_Network_Models_With_Application_to_Exchange-Rate_Data/links/00b7d5295fb1709e0a000000/Local-Stability-Analysis-of-Neural-Network-Models-With-Application-to-Exchange-Rate-Data.pdf

[2] Poggio T, Liao Q, Miranda B, Banburski A, Boix X, Hidary J. Theory IIIb: Generalization in deep networks. arXiv preprint arXiv:1806.11379. 2018 Jun 29.  https://arxiv.org/pdf/1806.11379.pdf


After reading the author's rebuttal, the reviewer's main concern has been met and the score of the paper has been raised.

**Time Spent Reviewing:**

4

---

> ### Author Response · Authors · 2021-08-10
> **We sincerely thank the reviewer for the helpful comments and suggestions! We provide a point-by-point response as follows.**
>
> 1). The optimization proposed in our paper is new. The mentioned two references did not conclude how to design a training procedure to make inputs reside around Lyapunov-stable hyperbolic equilibrium points.
>
> 2). For classification tasks, SODEF can be applied between the transformer encoder's output (corresponding to the classification token in the input sequence) and the classification head.
>
> 3).  Yes. In all experiments except the one shown in Fig. 4c, perturbations are added to the input $x$ by different attacks. Only in the one shown in Fig. 4c, we add perturbations to the features $\mathbf{z}(0)$ instead of the input $x$, which shows how much perturbation over feature $\mathbf{z}(0)$ the neural network can defend against.
>
> 4).  We mention that the norm used for all the attack experiments in the paper is $L_{\infty}$ in the supplementary material in lines 42-44 and we will explicitly include this parameter into Table 1 in the supplementary material. New PGD $\mathcal{L}_{\infty}$ attacks with much smaller steps is reported below as requested to the response for the next question. Additionally, as the reviewer requested for attack results for different norms, we have included here a thorough study with $\mathcal{L}_2$ attacks using the AutoAttack toolbox suggested. We use adversary.run\_standard\_evaluation to get the AutoAttack accuracy and use adversary.run\_standard\_evaluation\_individual to get each individual attack accuracy. Note that in the paper, for CIFAR10 and CIFAR100, the pixel values are in the interval (-2.429066, 2.753731) because of normalization. Here to test AutoAttack, we have strictly followed the instruction in \url{https://github.com/RobustBench/robustbench} to attack the original images before any normalization or resizing. (Additional results using AutoAttack with different norms are shown in Table 3. in the Response 1 to Reviewer 2. We refer the viewer to that response if interested.)
>
> In the first try, we only get accuracy under $10\%$ with integration time $T=5$ (same as the time set in the paper). As suggested in the discussion below Theorem 2 in the paper, if the malicious perturbations around the ODE input $\mathbf{z}(0)$ is not big, then the output $\mathbf{z}(T)$ for large enough $T$ will not be affected significantly by the perturbation. Since our Jacobian matrix is forced to have eigenvalues with negative real parts and has Lyapunov-stable property, we then increase the integration time which will mitigate the influence of the malicious perturbation. We include a detailed attack accuracy report for different times $T$.
>
> We test SODEF for CIFAR10 and CIFAR100 with $\epsilon=0.5$, same as in the AutoAttack paper. From the following  Table 5. and Table 6., we observe SODEF can defend FAB-attack and the black-box Square Attack even with small $T$. However the stronger two PGD type attacks beat SODEF with small $T$. The classification accuracy under different attacks all increases with increasing time $T$. For large $T$, SODEF is able to defend against all the four attacks included in AutoAttack.  We also test larger integration time $T>10$, but do not see any obvious improvements.
>
> |          Attack / $T$          |   1   |   3   |   5   |   6   |   7   |   8   |   9   |  10   |     |     |
> |:------------------------------:|:-----:|:-----:|:-----:|:-----:|:-----:|:-----:|:-----:|:-----:|:---:|:---:|
> |             Clean              | 95.10 | 94.95 | 95.05 | 95.00 | 94.92 | 95.10 | 95.11 | 95.02 |     |     |
> |       APGD$_{\text{CE}}$       | 3.52  | 8.20  | 8.59  | 12.59 | 73.05 | 92.41 | 92.58 | __92.67__ |     |     |
> | APGD$^{\text{T}}_{\text{DLR}}$ | 8.20  | 8.94  | 9.49  | 11.58 | 71.06 | __92.19__ | 91.88 | 91.80 |     |     |
> |        FAB$^{\text{T}}$        | 26.95 | 93.93 | 95.05 | 95.00 | 94.92 | 95.10 | __95.11__ | 95.02 |     |     |
> |             Squre              | 73.95 | 76.95 | 80.30 | 80.62 | 81.30 | 80.86 | 83.59 | __85.55__ |     |     |
> |           AutoAttack           | 0.10  | 2.73  | 3.34  | 4.52  | 24.22 | 80.16 | 81.25 | __82.81__ |     |     |
>
> __Table 5.__ Classification accuracy (%) under AutoAttack on adversarial CIFAR10 examples with $\mathcal{L}_2$ norm, $\epsilon=0.5$ and different integration time $T$ for SODEF.
>
>
> |          Attack / $T$          |   1   |   3   |   5   |   6   |   7   |   8   |   9   |  10   |     |     |
> |:------------------------------:|:-----:|:-----:|:-----:|:-----:|:-----:|:-----:|:-----:|:-----:|:---:|:---:|
> |             Clean              | 88.00 | 88.12 | 88.15 | 88.00 | 87.92 | 88.00 | 88.05 | 88.10 |     |     |
> |       APGD$_{\text{CE}}$       | 17.20 | 21.33 | 21.05 | 23.67 | 69.67 | 85.33 | __87.10__ | 86.88 |     |     |
> | APGD$^{\text{T}}_{\text{DLR}}$ | 21.02 | 21.00 | 22.00 | 26.00 | 63.30 | __86.90__ | 86.20 | 86.54 |     |     |
> |        FAB$^{\text{T}}$        | 86.33 | 85.10 | 86.36 | __87.70__ | 87.67 | 86.55 | 86.22 | 85.93 |     |     |
> |             Square             | 84.67 | 86.22 | 87.05 | 87.20 | 86.90 | 86.33 | __87.05__ | 86.75 |     |     |
> |           AutoAttack           | 2.00  | 3.53  | 4.87  | 4.33  | 30.66 | 78.80 | 78.97 | __79.10__ |     |     |
>
> __Table 6.__ Classification accuracy (%) under AutoAttack on adversarial CIFAR100 examples with $\mathcal{L}_2$ norm, $\epsilon=0.5$ and different integration time $T$ for SODEF.
>
> For a comparison, we also provide the results of applying AutoAttack to other baseline models mentioned in the paper. We set integration time $T=10$ for ODE, TisODE and SODEF.
>
>
> |         Attack / Model         | No ODE |  ODE  | TisODE | SODEF |     |     |
> |:------------------------------:|:------:|:-----:|:------:|:-----:|:---:|:---:|
> |             Clean              | 88.00  | 87.90 | 88.00  | 88.10 |     |     |
> |       APGD$_{\text{CE}}$       | 23.30  | 6.75  | 11.67  | __86.88__ |     |     |
> | APGD$^{\text{T}}_{\text{DLR}}$ |  7.33  | 22.00 | 22.42  | __86.54__ |     |     |
> |        FAB$^{\text{T}}$        | 79.30  | 78.67 | 79.50  | __85.93__ |     |     |
> |             Square             | 84.52  | 85.67 | 86.00  | __86.75__ |     |     |
> |           AutoAttack           |  0.00  | 1.33  |  2.52  | __79.10__ |     |     |
>
> __Table 7.__ Classification accuracy (%) under AutoAttack on adversarial CIFAR100 examples with $\mathcal{L}_2$ norm, $\epsilon=0.5$ and $T=10$.
>
> 5).  We do more experiments using step size $2.5*0.1/20$ while keeping all the other parameters the same as in Table 1 in the supplementary material and show the results in the following Table 8 and Table 9. PGD with smaller step size lowers the classification accuracy, however SODEF still outperforms the other baseline models. Additional AutoAttack experiment results are shown in the Response 4, where for larger integration times $T$, SODEF accuracy outperforms by a significant margin.
>
> |    Attack/Model    |     para.      | no ode | ODE  | TisODE | SODEF |     |     |
> |:------------------:|:--------------:|:------:|:----:|:------:|:-----:|:---:|:---:|
> |        None        |       \-       |  95.2  | 94.9 |  95.0  | 95.0  |     |     |
> |  step size $0.1$   | $\epsilon=0.1$ |  3.09  | 3.21 |  3.05  | __55.59__ |     |     |
> | step size $0.0125$ | $\epsilon=0.1$ |  0.91  | 0.84 |  1.03  | __52.45__ |     |     |
>
> __Table 8.__ Classification accuracy (%) on adversarial CIFAR10 examples under PGD attack with different step size.
>
> |    Attack/Model    |     para.      | no ode | ODE  | TisODE | SODEF |     |     |
> |:------------------:|:--------------:|:------:|:----:|:------:|:-----:|:---:|:---:|
> |        None        |       \-       |  88.3  | 88.1 |  87.9  | 88.0  |     |     |
> |  step size $0.1$   | $\epsilon=0.1$ |  2.39  | 3.36 |  3.44  | __22.35__ |     |     |
> | step size $0.0125$ | $\epsilon=0.1$ |  0.02  | 0.89 |  1.38  | __16.35__ |     |     |
>
> __Table 9.__ Classification accuracy (%) on adversarial CIFAR100 examples under PGD attack with different step size.
>
> 6). From Theorems 1 and 2, we see that a small perturbation around the Lyapunov-stable equilibrium point $\mathbf{z}(0)$ leads to $\tilde{\mathbf{z}}(t) \to \mathbf{z}(0)$ as $t\to \infty$, where $\tilde{\mathbf{z}}(t)$ is the ODE solution for the perturbed input $\tilde{\mathbf{z}}(0)$. That is why we force embedding features $\mathbf{z}(0)$ to locate near the Lyapunov-stable equilibrium points to ensure the robustness against small input perturbations. Moreover, it is undesirable to see Lyapunov-stable equilibrium points for different classes locating near each other because it would lead to poor adversarial defense. Therefore, we are only interested in those diversified Lyapunov-stable equilibrium points which are determined by our designed final FC layer.
>
> The robustness by forcing  $\mathbf{z}(0)$ in the stable neighborhood is also demonstrated in Response 1 to Reviewer 2 where in two experiments we fix the pretrained feature extractor $h_{ \phi }$  and append our SODEF block. Please refer to Table 2 and Table 3.
>
> 7). Yes, the adversarial examples are generated by attacking these three models independently.
>
> 8).  We will correct the typos. We include attack experiment with different attack parameters $\epsilon$ in Table 1. in the Response 1 to Reviewer 1. We refer the reviewer to see the results there.
>
> 9).  We have completed the Table 2 by giving results under PGD attack.
>
> | Attack |     Para.      |  ODE  | ODE$^+$ | TisODE | TisODE$^+$ |
> |:------:|:--------------:|:-----:|:-------:|:------:|:----------:|
> |  None  |       \-       | 99.6  |  99.7   |  99.6  |    99.7    |
> |  FGSM  | $\epsilon=0.3$ | 31.40 |  52.80  | 29.10  |   __63.50__    |
> |  PGD   | $\epsilon=0.3$ | 0.29  |  0.30   | 11.30  |   __20.20__    |
>
> __Table 10.__ Classification accuracy (%) on adversarial MNIST examples, where the superscript $^+$ indicates the last FC layer is fixed to be $\mathbf{V}$.

---

> > ### Comment · Reviewer_dTd1 · 2021-08-29
> > **Most concerns have been addressed and reviewer is happy to raise the score**
> >
> > > The mentioned two references did not conclude how to design a training procedure to make inputs reside around Lyapunov-stable hyperbolic equilibrium points.
> >
> > The reviewer agrees with this assessment and thanks the authors.
> >
> > >  Yes. In all experiments except the one shown in Fig. 4c, perturbations are added to the input  by different attacks. Only in the one shown in Fig. 4c, we add perturbations to the features  instead of the input , which shows how much perturbation over feature  the neural network can defend against.
> > > We mention that the norm used for all the attack experiments in the paper is  in the supplementary material in lines 42-44 and we will explicitly include this parameter into Table 1 in the supplementary material.
> > > Yes, the adversarial examples are generated by attacking these three models independently.
> >
> > Thanks for the clarifications. It will be good to include details in the final version.
> >
> > > New PGD  attacks with much smaller steps is reported below as requested to the response for the next question.
> >
> > The reviewer thanks the authors for doing this experimental evaluation.
> >
> > > Additional AutoAttack experiment results are shown in the Response 4, where for larger integration times , SODEF accuracy outperforms by a significant margin.
> >
> > The results from AutoAttack experiments are very convincing.

---

### Official Review · Reviewer_eqFz · 2021-07-16

**Rating:** 7
**Confidence:** 4

**Summary:**

This paper proposes a neural network classifier architecture based on neural ODEs that is designed to be robust to adversarial attacks. In the proposed method, the classes are represented by vectors, chosen to be maximally separated in the output space w.r.t. cosine similarity. The neural ODE system is then encouraged to have equilibrium points at those class vectors via regularization; the idea is that perturbed outputs are pulled towards the equilibria as the ODEs evolve.

**Limitations And Societal Impact:**

Yes

**Main Review:**

This paper builds on a line of work that connects neural ODEs with adversarial robustness. The approach was interesting - a diagonally dominant condition was used as a scalable approach to enforce stability conditions on the dynamics Jacobian. The various experiments visually and quantitatively show the potential benefits of locally attractive diverse class label targets. In my understanding, the main strength is that the SODEF layer mainly serves to stabilize the output of a pretrained classifier, and does not really require much predictive power of its own - this may make it easier to tune the several regularization hyperparameters that are required. It seems that a pretrained model can be used as the feature extractor $h_\phi$, but it is necessary that its output $z(0)$ is sufficiently close to the target diversified equilibrium points. Could a neural network layer have been used to map the output of a pretained model to a diversified $z(0)$, without needing to retrain the entire feature extractor? Also, Runge-Kutta order 5 was used to integrate SODEF. Was the proposed method compared with simpler ODE solvers such as forward Euler?

**Time Spent Reviewing:**

2

---

> ### Author Response · Authors · 2021-08-10
> **We sincerely thank the reviewer for the helpful comments and suggestions! We provide a point-by-point response as follows.**
>
> 1).  Your understanding is correct. To answer the first question regarding whether we need to train the feature extractor,  we performed two experiments here.
>
> (I). We first do an experiment for the CIFAR100 dataset where we fix the entire feature extractor (EffcientNet, pretrained on imagenet and CIFAR100, used in our paper already) except for the last layer in the extractor. The results are shown as follows in Table 2.
>
>
> |    Model / Attack     | clean | FGSM  |  PGD  |     |     |     |     |
> |:---------------------:|:-----:|:-----:|:-----:|:---:|:---:|:---:|:---:|
> | Fine-tuned (w/ SODEF) | 88.0  | __37.67__ | __22.35__ |     |     |     |     |
> |   Fixed (w/ SODEF)    | 87.1  | 33.65 | 17.32 |     |     |     |     |
> | Fixed (no ODE block)  | 88.0  | 25.32 | 2.39  |     |     |     |     |
>
> __Table 2.__ Classification accuracy (%) using SODEF under PGD and FGSM attack on adversarial CIFAR100 examples with parameters $\epsilon=0.1$.
>
> In the first line we use the pretrained EffcientNet with our SODEF and with fine-tuning. In the second line we use a fixed pretrained EffcientNet and with trainable SODEF. In the third line we just attack the pretrained EffcientNet (no ODE and no training since this opensource model has already been trained on CIFAR100 and get 88\% clean accuracy).
>
> We observe it performs worse than not fine-tuning a pretrained feature extractor (see also Table 2 in the supplementary material) but however still shows some stability than the vanilla model. One possible reason for the deterioration is the pretrained feature extractor does not have a well diversified $\mathbf{z}(0)$.
>
>
> (II). To show that our SODEF is compatible with many defense methods and can be applied to any neural network's final regressor layer to enhance its stability against adversarial attacks, we also include one robust network, TRADES [R2.1], as our feature extractor. The pretrained model is provided here https://github.com/P2333/Bag-of-Tricks-for-AT, and we choose the model with architecture "WRN-34-10" to conduct our experiments.  We use a very strong attack toolbox AutoAttack [R2.2] which is suggested by Viewer 3 and contains 4 different attacks.
>
> In this experiment we fix the pretained TRADES model (except the final FC layer (size 640x10) which is substituted by two trainable FC layers) as our feature extractor $h_{ \phi }$. We then append our (trainable) SODEF with integration time $T=5$ to the feature extractor. We attack the models using both the $L_2$ norm ($\epsilon=0.5$) and   $\mathcal{L}_{\infty}$ norm ($\epsilon=8/255$). The results are shown in Table 3. We clearly observe that our SODEF can enhance TRADES's robustness.
>
> |         Attack / Model         | TRADES $\mathcal{L}_{\infty}$ | TRADES+SODEF $\mathcal{L}_{\infty}$ | TRADES $\mathcal{L}_2$ | TRADES+SODEF $\mathcal{L}_2$ |     |     |
> |:------------------------------:|:-----------------------:|:-----------------------------:|:----------------:|:----------------------:|:---:|:---:|
> |             Clean              |          85.48          |             85.18             |      85.48       |         85.18          |     |     |
> |       APGD$_{\text{CE}}$       |          56.08          |             __70.90__             |      61.74       |         __74.35__          |     |     |
> | APGD$^{\text{T}}_{\text{DLR}}$ |          53.70          |            __64.15__            |      59.22       |         __68.55__          |     |     |
> |        FAB$^{\text{T}}$        |          54.18          |             __82.92__             |      60.31       |         __83.15__          |     |     |
> |             Square             |          59.12          |             __62.21__             |      72.65       |         __76.02__          |     |     |
> |           AutoAttack           |          53.69          |             __57.76__             |      59.42       |         __67.75__          |     |     |
>
> __Table 3.__ Classification accuracy (%) using TRADES (w/ and w/o SODEF) under AutoAttack on adversarial CIFAR10 examples with $L_2$ norm ($\epsilon=0.5$) and $\mathcal{L}_{\infty}$ norm ($\epsilon=8/255$).
>
>
>
> [R2.1] Hongyang Zhang, Yaodong Yu, Jiantao Jiao, Eric P Xing, Laurent El Ghaoui, and Michael I Jordan. Theoretically principled trade-off between robustness and accuracy. In International Conference on Machine Learning (ICML), 2019.
>
> [R2.2] Carlini N, Wagner D. Towards evaluating the robustness of neural networks. In 2017 ieee symposium on security and privacy (sp) 2017 May 22 (pp. 39-57). IEEE. https://arxiv.org/abs/1608.04644
>
>
> 2).   We test different ODE solvers including Runge-Kutta of order 5, Runge-Kutta of order 2, Euler method, Midpoint method, and Fourth-order Runge-Kutta with 3/8 rule. All the ODE solvers tested in Table 4. show similar performance. For Euler method, Midpoint method, and Fourth-order Runge-Kutta with 3/8 rule, we set step size to $0.05$. See https://github.com/rtqichen/torchdiffeq.
>
> |             Model / Attack             | clean | FGSM  |     |     |     |     |     |
> |:--------------------------------------:|:-----:|:-----:|:---:|:---:|:---:|:---:|:---:|
> |         Runge-Kutta of order 5         | 95.00 | 68.05 |     |     |     |     |     |
> |         Runge-Kutta of order 2         | 95.12 | 69.95 |     |     |     |     |     |
> |              Euler method              | 95.20 | 70.22 |     |     |     |     |     |
> |            Midpoint method             | 95.10 | 70.86 |     |     |     |     |     |
> | Fourth-order Runge-Kutta with 3/8 rule | 95.07 | 69.50 |     |     |     |     |     |
>
> __Table 4.__ Classification accuracy (%) using SODEF under FGSM attack on adversarial CIFAR10 examples with parameters $\epsilon=0.1$ and different ODE solvers.

---

> > ### Comment · Reviewer_eqFz · 2021-09-01
> > **Thank you for author response**
> >
> > The additional experiments are thorough and provide helpful context for better understanding the proposed methods. My main concerns have been addressed, and I would like to increase my score to a 7 (accept).

---

### Official Review · Reviewer_8u2W · 2021-07-17

**Rating:** 8
**Confidence:** 5

**Summary:**

This paper tries to improve the robustness of neural ODE via enforcing the Lyapunov stability constraints in the training process. The key idea is as follows: a dynamic system can have multiple equilibrium points, and the input perturbation around the equilibrium point will decay and not significantly influence the result if the equilibrium point is Lyapunov stable. This is an excellent contribution that connects the Lyapunov stability of ODEs with the robustness of neural networks.

**Limitations And Societal Impact:**

It would be great if the following points can be discussed/clarified:
1. The linearization analysis only works if the perturbation is small. How strong is this assumption?
2. How many equilibrium points should the system have in the training process? How can we control the number of equilibrium points? What's the relationship of the number of equilibrium points with the number of output labels?

A potential limitation: this work only compared with standard neural ODE, and has not been compared with other robust neural ODE, for instance:
--Chen Z, Li Q, Zhang Z. Towards Robust Neural Networks via Close-loop Control. In International Conference on Learning Representations 2020 Sep 28
--Dinghuai Zhang, Tianyuan Zhang, Yiping Lu, Zhanxing Zhu, and Bin Dong.  You only propagate once: Accelerating adversarial training via maximal principle. In Advances in Neural Information Processing Systems, pp. 227–238, 2019.

**Main Review:**

This is a very well written paper. The authors proposed a new neural network to enforce the robustness of neural networks. The network consists of three blocks: (1) a feature extractor block, (2) a neural ODE block that satisfies the Lyapunov stability condition, and (3) a diversity promoting FC layer. By enforcing the Lyapunov condition of the neural ODE, the input perturbation is guaranteed to decay and will not significantly influence the prediction.

On the implementation side, the Lyapunov condition is achieved by enforcing the Jacobian matrix A to be dominantly diagonal and have eigen values with negative real parts. These constraints appear in the regularization term of the training loss function.

The research results are well justified by theoretical analysis. The proposed Lyapunov stable model has shown significant robustness improvement compared with neural ODE trained without Lyapunov stability constraints.

Some minor comments:
1. Some important references of neural ODE are missing. The idea of neural ODEs were firstly proposed in the applied math community, and the following papers should be cited:
--W. E,  A proposal on machine learning via dynamical systems. Communications in Mathematics and Statistics, 5(1):1–11, 2017.
--Eldad Haber and Lars Ruthotto.  Stable architectures for deep neural networks. Inverse Problems,34(1):014004, 2017.

2, The following papers improve the robustness of neural ODEs via open-loop and close-loop controls. They are ignored in this paper:
--Chen Z, Li Q, Zhang Z. Towards Robust Neural Networks via Close-loop Control. In International Conference on Learning Representations 2020 Sep 28
--Dinghuai Zhang, Tianyuan Zhang, Yiping Lu, Zhanxing Zhu, and Bin Dong.  You only propagate once: Accelerating adversarial training via maximal principle. In Advances in Neural Information Processing Systems, pp. 227–238, 2019.

3. The authors assume that the ODE is time-invariant. I wonder how they handle the model parameters (e.g., weight matrices and convolution filters) at different layers: are they the same across different layers?

4. More details should be provided to explain why we need the feature extraction block and how this block should be designed for general cases.



**Time Spent Reviewing:**

3 hours

---

> ### Author Response · Authors · 2021-08-10
> **We sincerely thank the reviewer for the helpful comments and suggestions! We provide a point-by-point response as follows.**
>
> 1). Thank you for the references mentioned in 1 and 2. We will include them in the revision. A neural ODE network can be set to time-invariant with no input of time $t$ to the ODE block. Roughly speaking, in the literature, time-variant ODE takes time $t$ as one (or more) dimension if the ODE network is composed of multiple FC layers and takes time $t$ as one channel if the ODE network is composed of convolutional layers. Time-invariant ODE network does not have such time input. In our experiments, time-invariant ODE consists of just the regular FC or convolutional layers. The weight matrices are not kept to be the same across different FC layers while convolution filters in our experiment are set to have the same dimension across different layers for simplicity. In both cases we need to keep the input dimension and output dimension the same.
>
> 2). The constraints (3) to (5) in the paper force $\mathbf{z}(0)$ extracted from the feature extraction block to be near the Lyapunov-stable equilibrium points with strictly diagonally dominant derivatives. Theoretically speaking, any other general network blocks can be used before the ODE block $f_{\theta}$ since our objective function in (2)-(6) is general enough to handle any such general network blocks. We however suggest to put at least one layer of trainable network before the ODE block instead of directly applying inputs to it. This is because with no preceding blocks, the constraints (3) to (5) will in principle force the ODE block $f_{\theta}$ to adjust its trainable parameters with the goal of making the Jacobian matrix evaluated at every single input in the training set to have negative real parts, which is a challenging task for the ODE block. We therefore use some feature extraction block $h_{ \phi }$ to map the inputs of each class to some representative high dimensional space with our diversity promoting technique introduced in Section 3.1. This makes the optimization with constraints (3) to (5) easier in practice.
>
> 3).  We note that in adversarial attacks, perturbations are assumed to be imperceptible [R1.1], otherwise attack detection techniques can be used [R1.2]. Therefore, most related literature on robust techniques against adversarial attacks assumes small perturbation. However, it is hard to obtain a theoretical analysis for an exact perturbation bound, which will highly depend on specific datasets and model parameters. We instead demonstrate it experimentally. The result shown in Fig.~4c shows how the accuracy changes when we increase the FGSM attack strength $\epsilon$. In that experiment, we directly perturb the feature $\mathbf{z}(0)$ instead of the input $x$, which shows how much perturbation over feature $\mathbf{z}(0)$ the neural network can defend against. Since perturbation in the input $x$ is altered by the feature extractor $h_{ \phi }$, to show how much perturbation over $x$ the neural network can defend against, we additionally include one experiment here for SODEF where we increase the FGSM attack strength $\epsilon$ at input $x$. All parameters used here, except $\epsilon$, are the same as in Table 1 in the supplementary material. Note that in the paper, for CIFAR10 and CIFAR100, the pixel values are in the interval $(-2.429066, 2.753731)$ because of the normalization we have used.  The results are shown in Table 1. As expected, when the perturbation is smaller, SODEF can provide good adversarial robustness.
>
> [R1.1] Szegedy, C.; Zaremba, W.; Sutskever, I.; Bruna, J.; Erhan, D.; Goodfellow, I.; and Fergus, R. 2013. Intriguing properties of neural networks. In Proc. Int. Conf. Learning Representations.
>
> [R1.2] Meng, Dongyu, and Hao Chen. "Magnet: a two-pronged defense against adversarial examples." Proceedings of the 2017 ACM SIGSAC conference on computer and communications security. 2017.
>
> | Dataset / $\epsilon$ | clean | 0.01  | 0.02  | 0.04  | 0.06  | 0.08  | 0.10  | 0.15  | 0.30  | 0.50  | 1.00  | 2.00  |     |     |     |
> |:--------------------:|:-----:|:-----:|:-----:|:-----:|:-----:|:-----:|:-----:|:-----:|:-----:|:-----:|:-----:|:-----:|:---:|:---:|:---:|
> |       CIFAR10        | 95.00 | 91.03 | 87.03 | 81.42 | 78.05 | 72.80 | 68.25 | 62.58 | 46.16 | 27.49 | 17.56 | 10.63 |     |     |     |
> |       CIFAR100       | 88.00 | 66.15 | 62.86 | 59.10 | 54.45 | 46.71 | 37.24 | 18.09 | 5.90  | 3.20  | 1.09  | 0.89  |     |     |     |
>
> __Table 1.__ Classification accuracy (%) using SODEF under FGSM attack on adversarial CIFAR10 and CIFAR100 examples with different parameters $\epsilon$.
>
> 4).  In Theorem 4, we showed that there exist functions in this space such that each support $E_l$ for each class $l$ contains at least one Lyapunov-stable equilibrium point. There may exist more than one equilibrium point for each class. In our current work, we cannot control the exact number of equilibrium points for each class. However, to promote the number of equilibrium points, we have included $\sin$ as our activation function in experiments since the $\sin$ curve has many zero equilibrium points and potentially leads to more Lyapunov-stable equilibrium points.
>
> 5). Thanks for enlightening us with these two interesting works. Both of them recast training or inference problem as a discrete time differential game which can be solving using Pontryagin’s Maximum Principle whereas our focus is on ensuring the stability of continuous ODE neural network. Here are some fundamental differences between these two works and ours:
>
>     (1) Close-loop control neural network (CLC-NN) (Chen et al., 2019) is an inference algorithm using close-loop control for a given trained neural network (while ours is a training algorithm). When performing inference for a trained $T$-layer neural network, CLC-NN adds an additional control block (e.g., an auto-encoder which is trained offline using training data) to each layer neural network so CLC-NN actually explores discrete "layer-by-layer" dynamics rather than continuous dynamics.
>
>     (2) (Chen et al., 2020) focused on accelerating adversarial training by recasting the adversarial training problem as a discrete time differential game and it splits the adversary computation and weight updating where the adversary computation is only performed on the first layer (due to some observations from Pontryagin’s Maximum Principle). Our work pursues different direction to improve robustness, i.e., imposing the stability conditions on ODE, and we do not require adversarial training.
>
> We would consider these two works as cooperators (cooperating with our method may further improve the robustness) rather than competitors. For more numerical comparisons, please refer to Tables 2, 3 and 4 in our response to reviewer 2 and Tables 5-9 in our response to reviewer 3.

---

> > ### Comment · Reviewer_8u2W · 2021-08-30
> > **Main issues are clarified.**
> >
> > Thanks for the authors' detailed clarifications.
> >
> > Most of my comments have been addressed, including the missing references, the details of time-variant ODEs, and details of the feature extraction block.
> >
> > My question about the linearization analysis was not fully answered. However, I understand that this is a challenging question, and it may be worth future investigation.

---

### Author Response · Authors · 2021-08-26
**To all reviewers**

In the rebuttal, we have conducted extensive additional experiments according to reviewers' requests to demonstrate the efficiency of our work. We will include all the experiments in the rebuttal to the paper if it gets accepted. We sincerely thank all the reviewers for their valuable suggestions or comments for the experiments, which have improved the demonstration of the efficacy of the proposed SODEF model.

Lastly, since all comments here are public, we would like to sincerely thank the TisODE authors for providing the source codes in this crucial rebuttal period.

---

### Decision · Program_Chairs · 2021-09-27

**Decision:**

Accept (Poster)

**Comment:**

This paper proposes a neural network classifier architecture based on neural ODEs against adversarial attacks. Some of the reviewers have concerns on the experiments, while in the rebuttal the newly added experiments convince the reviewers. All reviewers finally give positive support to the paper. Thus, I recommend accepting the paper and the authors should include the new experiments in the final version.